# Expression variation and covariation impair analog and enable binary signaling control

Kyle M Kovary, Brooks Taylor, Michael L Zhao & Mary N Teruel* [iD]

## Abstract

Due to noise in the synthesis and degradation of proteins, the concentrations of individual vertebrate signaling proteins were estimated to vary with a coefficient of variation (CV) of approximately 25% between cells. Such high variation is beneficial for population-level regulation of cell functions but abolishes accurate single-cell signal transmission. Here, we measure cell-to-cell variability of relative protein abundance using quantitative proteomics of individual *Xenopus laevis* eggs and cultured human cells and show that variation is typically much lower, in the range of 5–15%, compatible with accurate single-cell transmission. Focusing on bimodal ERK signaling, we show that variation and covariation in MEK and ERK expression improves controllability of the percentage of activated cells, demonstrating how variation and covariation in expression enables population-level control of binary cell-fate decisions. Together, our study argues for a control principle whereby low expression variation enables accurate control of analog single-cell signaling, while increased variation, covariation, and numbers of pathway components are required to widen the stimulus range over which external inputs regulate binary cell activation to enable precise control of the fraction of activated cells in a population.

**Keywords** cellular heterogeneity; MAPK/MEK/ERK signaling; selected reaction monitoring mass spectrometry; SRM-MS; single-cell proteomics

**Subject Categories** Quantitative Biology & Dynamical Systems; Signal Transduction; Transcription

**Mol Syst Biol. (2018) 14: e7997**

## Introduction

Vertebrate signaling has been shown to control both binary and analog outputs. Here, we use the term binary if the output is bimodal and the term analog if the output signal changes in parallel with the input signal without bifurcations during the transmission. Examples of binary signaling decisions include the commitment to start the cell cycle (Cappell *et al*, 2016), cell differentiation (Chang *et al*, 2008; Jukam & Desplan, 2010; Ahrends *et al*, 2014), apoptosis (Spencer *et al*, 2009), action potentials (Hodgkin & Huxley, 1952) and the explosive secretory response of mast cells when encountering an antigen (Hide *et al*, 1993). Effective analog signaling in individual cells has been observed, for example, in the visual transduction system where the number of absorbed photons proportionally increases electric outputs in cone cells (Arshavsky *et al*, 2002), in single-cell IP3 and $Ca^{2+}$ regulation by GPCRs (Nash *et al*, 2001), as well as for CD-8 (Tkach *et al*, 2014) and IL-2 signaling (Feinerman *et al*, 2008) in T cells. Analog signaling is also needed to accurately regulate the timing or duration of intermediate cell processes such as in the cell cycle where the time between the start of S-phase to mitosis has only small variation between individual cells (Spencer *et al*, 2013). Such precise regulation of durations requires low noise in the signaling steps before mitosis (Kar *et al*, 2009). Together, these examples suggest that accurate analog signaling is important for graded control of cell outputs in single cells as well as for accurate internal timing.

A main motivation for our study was the high levels of protein expression variation that have been reported in vertebrate cells with coefficient of variations (CVs) of approximately 25% (Sigal *et al*, 2006; Spencer *et al*, 2009; Gaudet *et al*, 2012). Such high levels of expression variation are beneficial for binary signaling which is often regulated at the population level rather than single-cell level. In population-based signaling, a goal of organisms is to use different levels of input to regulate the percentage of cells in a population that make a binary decision such as whether to proliferate, differentiate, or secrete. For input stimuli to control which percentage of cells are activated, high noise in signaling is needed between cells in the population such that individual cells have different sensitivities to input stimuli (Süel *et al*, 2007; Raj & van Oudenaarden, 2008; Kalmar *et al*, 2009; Eldar & Elowitz, 2010; Ahrends *et al*, 2014). However, the same high noise needed to control population-level signaling does not have any benefit for analog signaling and just serves to degrade signal transmission. These different demands on noise for analog and binary signaling suggest that there is a trade-off for noise between population-level and single-cell signaling (Suderman *et al*, 2017). Specifically, the reported high levels of expression variation and signaling noise in mammalian cells (Sigal *et al*, 2006; Cheong *et al*, 2011; Gaudet *et al*, 2012; Selimkhanov *et al*, 2014) raise the question of how noise in a signaling system can be low enough for accurate analog signaling. It also remained unclear how the different potential internal noise sources could

Department of Chemical and Systems Biology, Stanford University, Stanford, CA, USA
*Corresponding author. Tel: +1 650 721 2045; E-mail: mteruel@stanford.edu

generate optimal conditions for analog single-cell versus binary population-level signaling.

Here, we measure cell-to-cell variation in the relative abundance of pathway components to understand the limits of analog and binary signaling accuracy. We also investigated the role of covariation of pathway components since we realized that covariation could exacerbate the analog signaling problem and/or enable the control of population-level binary signaling. We considered that previous estimates of cell-to-cell variation in protein expression might be too high due to experimental challenges in accurately measuring small differences in protein abundance between cells and accounting for "hidden variables" such as differences in cell size and cell cycle state (Symmons & Raj, 2016). To determine lower limits of protein variation, we developed single-cell quantitative proteomics methods in single *Xenopus laevis* eggs and employed quantitative normalization of cultured human cells to accurately measure variations in protein abundance normalized by protein mass. We found that cell-to-cell variation in relative protein abundance is much lower than expected, with CVs of between 5 and 15%, suggesting that expression variation is less limiting than currently believed and is compatible with accurate analog signal transmission. Furthermore, our simulations show that these experimentally observed low levels of expression variation pose a challenge for cells to accurately control population-level decisions. One potential strategy to increase pathway output variation was revealed by experiments which showed significant covariation between the single-cell expression of two sequential signaling components, MEK and ERK. Our modeling showed that such increased covariation—which increases the overall noise in the signaling pathway—allows populations of cells to control the percentage of cells that activate ERK over a wider range of input stimuli, suggesting that covariation of signaling components is one strategy for populations of cells to more accurately control binary cell-fate decisions. Finally, we developed a metric to describe how systems can optimize the shared use of pathway components to control single-cell analog and population-level binary signal transmission by using different numbers of regulatory components, levels of expression variation, and degrees of covariation.

## Results

### Computational simulations using reported levels of expression variation show a dramatic loss of analog single-cell transmission accuracy

Our study was motivated by the reported high levels of expression variation and the detrimental impact that this source of noise may have on analog single-cell signaling, especially since signaling pathways typically have multiple components which necessarily results in even higher cumulative signaling noise. To define the general control problem of how expression variation increases overall signaling noise and limits signaling output accuracy, we carried out simulations by applying a relative fold-change in input signal (R) to a signaling pathway and stochastically varying the expression of pathway components for each simulation. To determine how accurately a multi-step signaling pathway can transmit a relative input stimulus (R) to an analog output (A*), we modeled the signaling

pathway shown in Fig 1A. Specifically, we used a five-step model where a relative change in input R acts through four intermediate steps, possibly reflecting a kinase cascade with counteracting phosphatases, to generate corresponding changes in the output A*. The regulation of these steps can be at the level of activity or localization of pathway components. We considered five steps with 10 variable regulators to be a typical signaling pathway since it has been shown that step numbers in signaling pathways can range from very few in visual signal transduction (Stryer, 1991) to over 10 steps in the growth-factor control of ERK kinase and cell cycle entry (Johnson & Lapadat, 2002). In our simulations, each of the parameters represents a regulatory protein that activates or inactivates one of the pathway steps. We assumed that each of these components has "expression variation," meaning that their concentrations vary between cells with a coefficient of variation (CV) calculated as their standard deviation divided by their mean value in the cell population. We simulated this expression variation by multiplying each parameter in the model with a lognormal stochastic noise term with a CV of either 5, 10, or 25% (Ahrends *et al*, 2014). As is apparent in the top plots in Fig 1B for a CV of 5%, the signaling responses of cells to threefold (red) and ninefold (blue) increases in the input stimulus, R, can be readily distinguished from the signaling responses of unstimulated cells (black traces). For a higher CV of 10%, the signaling responses to a threefold increase in R partially overlap with the unstimulated cell responses, and only the responses to a ninefold increase in R can be unequivocally distinguished from unstimulated cell responses. For a CV of 25%, even responses to a ninefold increase in input stimulus overlap with the responses of unstimulated cells, showing a dramatic loss in signaling accuracy.

One way to overcome this dramatic loss in signaling accuracy due to expression variation of pathway components is to increase the input stimulus. We reasoned that we could use a fold-increase parameter to quantify the loss in signal accuracy. We thus defined a fold-Input Detection Limit (fIDL) as the minimal fold-stimulus needed to generate signaling responses that can, in 95% of cases, be distinguished from cell responses in unstimulated cells (see Materials and Methods for calculation). Figure 1C shows an example of how the fIDL is calculated by determining the minimum fold-input stimulus that is needed to have only a 5% overlap between the resulting signaling output distributions (A*) of unstimulated and stimulated cells (black and green histograms, respectively). In the case shown, an fIDL stimulus of 2.83 is needed to overcome the loss of signaling accuracy caused by having 10% expression variation in pathway components. We used fIDL instead of a commonly used mutual information metric since mutual information between input (R) and output (A*) has a strong dependency on the dynamic range of the system output, while the fIDL is largely independent of saturation (Fig EV1). As shown in the barplots in Fig 1D, increasing the CV of pathway components from 10 to 25% increases the fIDL from 2.83 to 14, a stimulus requirement that is likely prohibitive for analog single-cell signal transmission. Our realization that fIDLs are very high for reported expression variation levels was a main motivation for our strategy below to more accurately measure expression variation in order to understand whether and how analog signaling in single cells is limited by this noise source.

We also wanted to determine whether the expression of vertebrate proteins may covary since covariance has been shown to exist

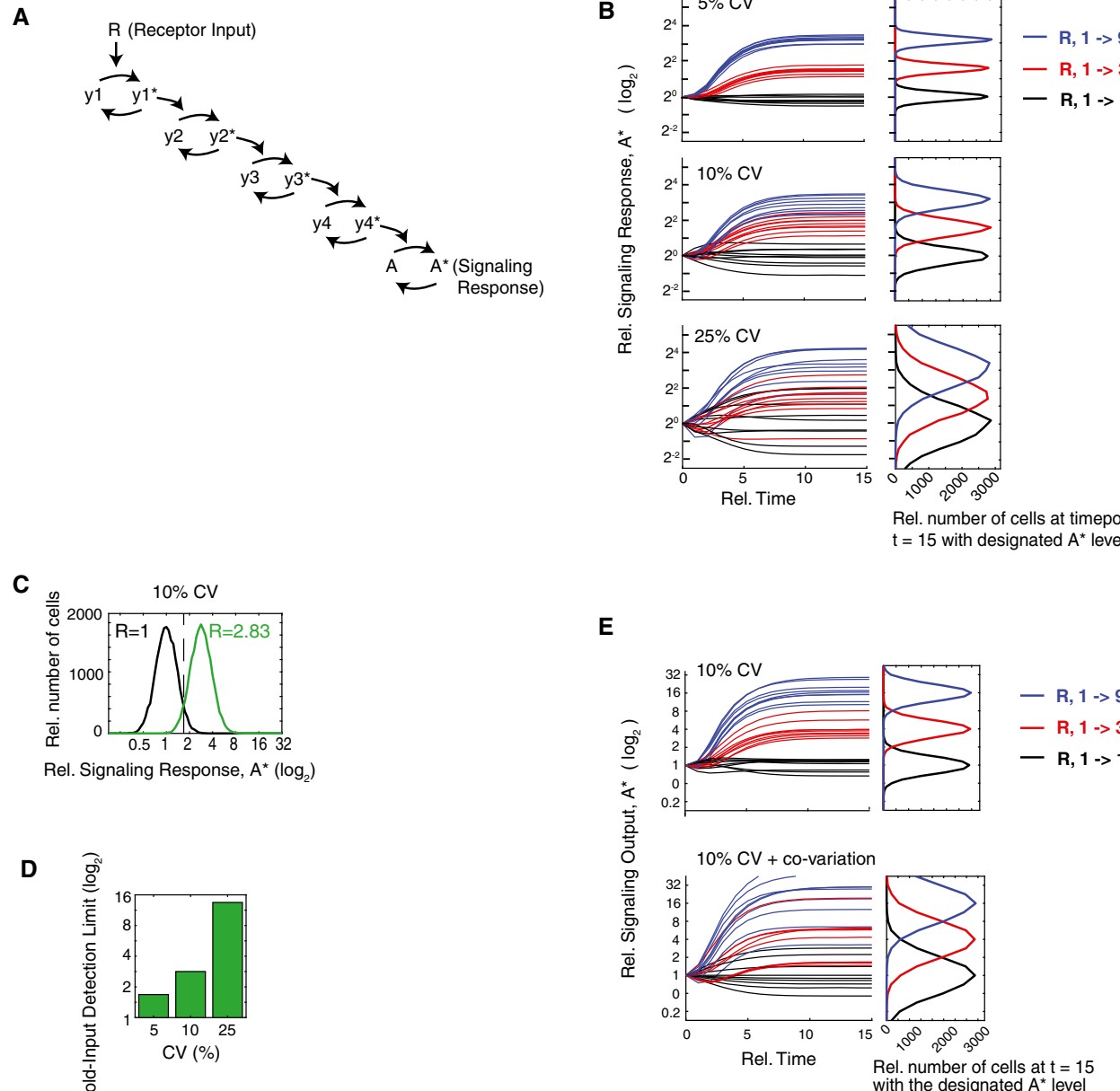

**Figure 1.  Computational simulations using reported levels of expression variation show a dramatic loss of analog single-cell transmission accuracy.**

A   Schematic of a five-step analog signaling pathway where the asterisk (*) represents the activated form which is assumed in this model to be a small fraction of the total.

B   The timecourse plots show how relative threefold (red) and ninefold (blue) input changes in R result in analog output responses with different degrees of noise. Random lognormal expression variation was added simultaneously to each pathway component. The accuracy of analog signal transmission is dramatically reduced as the coefficient of variations (CVs) increase from 5% (top), 10% (middle), to 25% (bottom).

C   Example of the output response distributions of unstimulated (black) and stimulated (green) cells at the fold-Input Detection Limit (fIDL). The fIDL represents the minimal stimulus, R, needed to distinguish the output of stimulated cells from unstimulated cells with 95% accuracy, as marked by the vertical black dashed line. For the system in (A) with a 10% CV in each pathway component, the fIDL is 2.83.

D   Barplot comparing the fIDL values for the system in (A) with CVs of 5, 10, and 25%.

E   Simulation of the pathway model in (A) but now comparing the situation in which the pathway components are all uncorrelated with each other (top) with the situation in which the activating and de-activating pathway components covary with each other, respectively (bottom). The overlapping output distributions in the right panels show that covariance of components in the same pathway would introduce a marked loss in signal transmission accuracy.

in a yeast regulatory pathway (Stewart-Ornstein *et al*, 2012). We considered that if proteins within a signaling pathway covary, the overall noise in the output response would increase. To illustrate

the effect of covariance can have on a multi-step analog signaling pathway, we added covariation to the model shown in Fig 1A by making the positive regulators (e.g., kinases) covary together and

also made the negative regulators (e.g., phosphatases) covary together. As shown in Fig 1E, covariance causes the error propagation to increase, and the overall noise in the signaling output is much higher compared to the case where proteins in the same pathway vary independently of each other. Given that covariation causes a marked increase in the overall noise of the signaling response, one would expect that covariation between components of the same signaling pathway should generally be avoided in order to have accurate analog signaling.

### Development of a method to accurately measure the relative abundance of tens of proteins in a single cell

To probe the lower limits of protein expression variation, we selected a system with a need for analog single-cell signaling that was also suitable for parallel proteomics analysis. We chose *Xenopus laevis* eggs for three reasons. First, previous studies showed that the timing of the cell cycle during early embryogenesis is very precise with an accuracy of ~5% (Tsai *et al*, 2014), suggesting that the *Xenopus* system must have accurate analog signaling to maintain such timing. Second, eggs do not grow in size and have only minimal new synthesis and degradation of mRNA, two features which we thought would reduce protein expression variation. Third, *Xenopus laevis* eggs are well suited for single-cell proteomics analysis due to their large size (Ferrell, 1999), allowing us sufficient starting material to very sensitively measure and compare relative abundances of many proteins simultaneously in the same cell.

To accurately compare the relative abundance of tens of endogenous proteins in parallel in single cells, we used selected reaction monitoring mass spectrometry (SRM-MS), a low-noise quantitative mass spectrometry method (Abell *et al*, 2011; Picotti & Aebersold, 2012; Ahrends *et al*, 2014) (Fig EV2). Cytoplasmic proteins were extracted from eggs and subjected to trypsin digestion and phosphatase treatment before undergoing targeted quantification on a triple quadrupole mass spectrometer. Heavy isotope-labeled reference peptides were spiked in proportionately to a measured total protein concentration, and the ratio of the light (endogenous) peptide to the heavy (synthetic) peptide was used as a readout of relative protein abundance. Small calibration errors were further corrected for during the analysis using the median of 22 normalized peptide intensities as a correction factor similar to previous studies (Abell *et al*, 2011; Ludwig *et al*, 2012; Picotti & Aebersold, 2012). We measured relative protein abundance (abundance over total protein mass) as a measure of protein concentration since reaction rates and signaling processes depend on the concentration rather than abundance of proteins (Padovan-Merhar *et al*, 2015).

We first validated our method using bulk cell analysis at different timepoints during the first cell cycle, a process which can be initiated by addition of calcium ionophore and takes approximately 90 min to complete (Rankin & Kirschner, 1997). We measured the abundances of a set of 26 proteins that we selected to include known regulators of signaling and cell cycle progression, as well as several control proteins (Fig 2A; Table EV1). Timecourse analysis over the first cell cycle further showed that we could observe the expected cycling behavior of Cyclin A and Cyclin B (Fig 2B). We next demonstrated that we could measure timecourses of relative protein abundances in single cells by carrying out measurements at five timepoints with five eggs each (Fig 2C; Table EV2). Except for a

few known cell cycle-regulated genes, Cyclin A, Cyclin B, Cdc6 and Emi1, all of the measured proteins changed their abundance on average less than a few percent during the first egg cell cycle (Peshkin *et al*, 2015). The constant average level of many of these signaling and cell cycle proteins can in part be explained by only minimal mRNA synthesis during early *Xenopus laevis* cell cycles (Krauchunas & Wolfner, 2013).

### Low variation in the relative abundance of proteins explains how cells are able to accurately control analog single-cell functions

We next focused on analyzing the extent to which protein concentrations vary between single cells. We first analyzed the set of 25 individual eggs from Fig 2C and determined the variation of each protein in each of the batches of five eggs collected at each of the five timepoints (Fig 3A, left). Markedly, all CVs were much lower than expected with the median CV across all proteins and timepoints being only 7% (Fig 3A, histogram in right panel). To independently verify these low variation measurements, we collected and analyzed a larger set of 120 individual eggs: 60 eggs collected at 60 and at 80 min after activation. To test for reproducibility of the measured variation, we divided the 60 eggs at each timepoint into batches and carried out a variation analysis (Fig 3B; Table EV3). Bootstrapping analysis showed similar low variation (Fig EV3). As further validation, the variations measured in the two independent experiments were similar to each other (Fig 3C). We also noted that most of the proteins that have high cell-to-cell variation (marked as red circles in Fig 3C) also change their abundance during the cell cycle (Fig 2C), suggesting that high CVs reflect proteins whose abundances are actively regulated. Thus, our finding of low CVs answers the question raised in Fig 1A–D of how cells can accurately control analog single-cell signaling outputs. Since expression variation can be as low as 5–10%, this main source of signaling noise is compatible with accurate single-cell signaling and timing control. Such low variation may also permit accurate timing in the *Xenopus laevis* embryonic cell cycle, which has been measured to be on the order of ± 5% between eggs (Tsai *et al*, 2014).

It should be noted that for some proteins, the biological variation might be even lower than we were able to measure in these experiments. To test whether there is a lower limit for measuring variation, we carried out control experiments in which 30 individual eggs were lysed and mixed together to remove biological variability (Fig EV4A). This mixed lysate was then pipetted into 30 individual tubes, and the sample in each tube was prepared and analyzed separately by SRM mass spectrometry. The variation between these 30 individually prepared and analyzed aliquots of the same starting lysate were compared to obtain a measure of technical variation. As shown in Fig EV4B, the technical variation is comparable to the lowest CV measurements we show in Fig 3A–C, suggesting that further technical improvements may reveal even lower biological variation.

Our analysis so far argues that expression variation can be much lower than previously assumed, which would enable accurate analog single-cell signaling as shown by how decreasing expression variation in Fig 1B allows for less overlap between unstimulated and stimulated cell responses. We next tested whether we would find the same low variation in protein expression in cultured human cells (HeLa cells) by carrying out immunocytochemistry

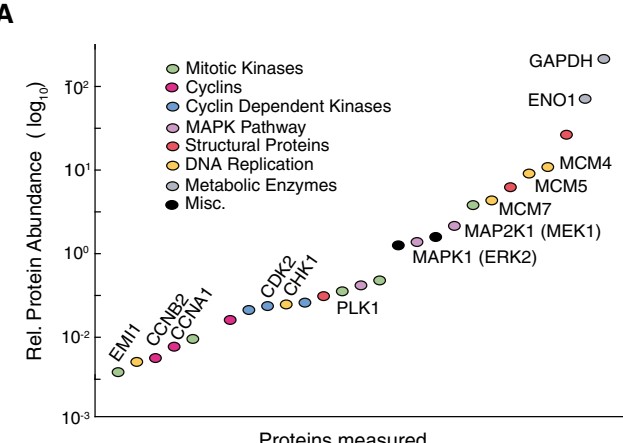

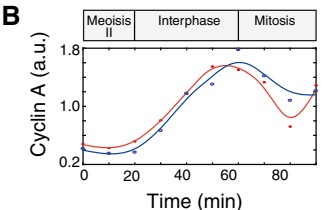

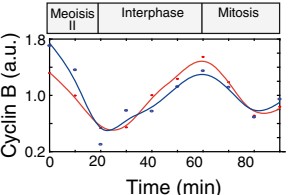

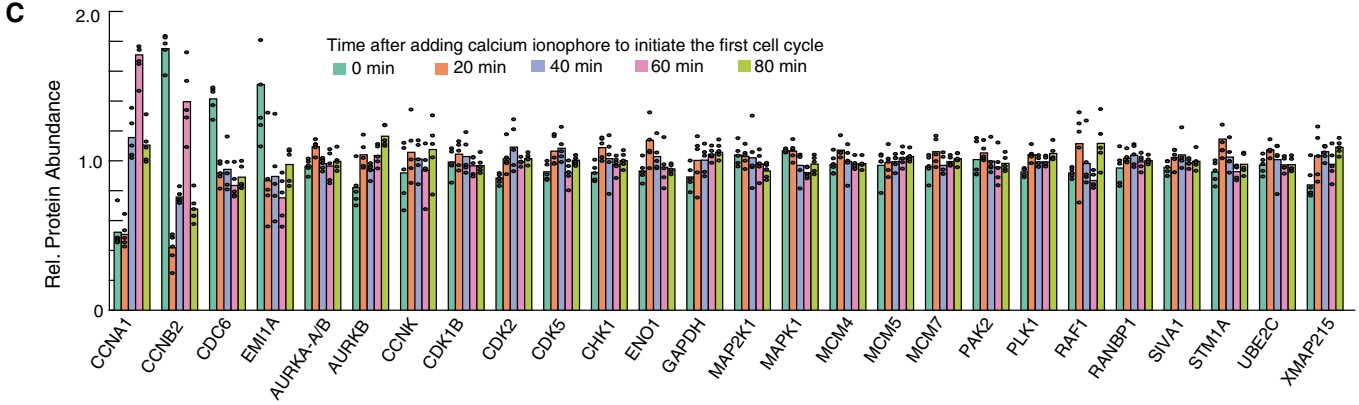

**Figure 2.   Development of a method to quantitatively measure relative abundances of tens of endogenous proteins in parallel in single *Xenopus* eggs.**

A   Comparison of protein abundance of a set of cell cycle, signaling and control proteins in *Xenopus* eggs. Abundance measurements are based on SRM-MS measurements of the combined cell extracts from 5 eggs collected at the same time and before initiation of the first cell cycle. Quantitation of relative protein abundance was carried out by adding heavy isotope-labeled reference peptides to the egg extracts.

B   Timecourse analysis of changes in Cyclin A and Cyclin B levels during the first *Xenopus* cell cycle measured in combined cell extracts from 5 eggs per timepoint.

C   Five individual eggs were collected at five timepoints: 0, 20, 40, 60, and 80 min after the addition of calcium ionophore. To minimize variability due to sample handling and instrument sources, the 25 individual eggs were prepared for mass spectrometry analysis at the same time and were then analyzed in sequential runs on the same mass spectrometer. Barplot shows relative abundance changes of the 26 proteins shown in (A) tracked through the first egg cell cycle. Each black dot represents the value from an individual egg.

experiments (Figs 3D and EV5). To accurately measure relative protein abundances, we first gated for cells in the same G0/G1 cell cycle state by using Hoechst DNA stain measurements (2n-peak; Cappell *et al*, 2016). We further normalized the abundance of each protein to total protein mass in each cell. The latter was measured using an amine-reactive dye that stains all proteins in a cell (Kafri *et al*, 2013). Since total protein mass is proportional to cell volume (Grover *et al*, 2011), normalization by total protein mass can be used as a measure of protein concentration, analogous to the normalization we used in the single-egg experiments. To minimize small illumination non-uniformities associated with imaging, we also confined our analysis to cells in the center area of images where the illumination and light collection is more uniform (see Materials and Methods). For comparison with the *Xenopus* egg data, we measured corrected CVs for the relative abundances for ERK, MEK,

MCM5, and MCM7 as well as the control proteins GAPDH and ENO1. We validated the specificity of the antibodies by showing that the immunocytochemistry staining could be knocked down by the respective siRNAs (Fig EV6). The resulting CVs for relative protein abundance were in the 10–15% range, lower than typically reported mammalian protein CV values (Sigal *et al*, 2006; Niepel *et al*, 2009; Gaudet *et al*, 2012).

## Identification of covariance between the relative abundances of components in the ERK pathway

We next determined whether there was covariance between proteins by analyzing the same 120-egg proteomic dataset shown in Fig 3B. As shown in Fig 4A, our correlation analysis uncovered several covarying regulatory proteins. For example, there was significant

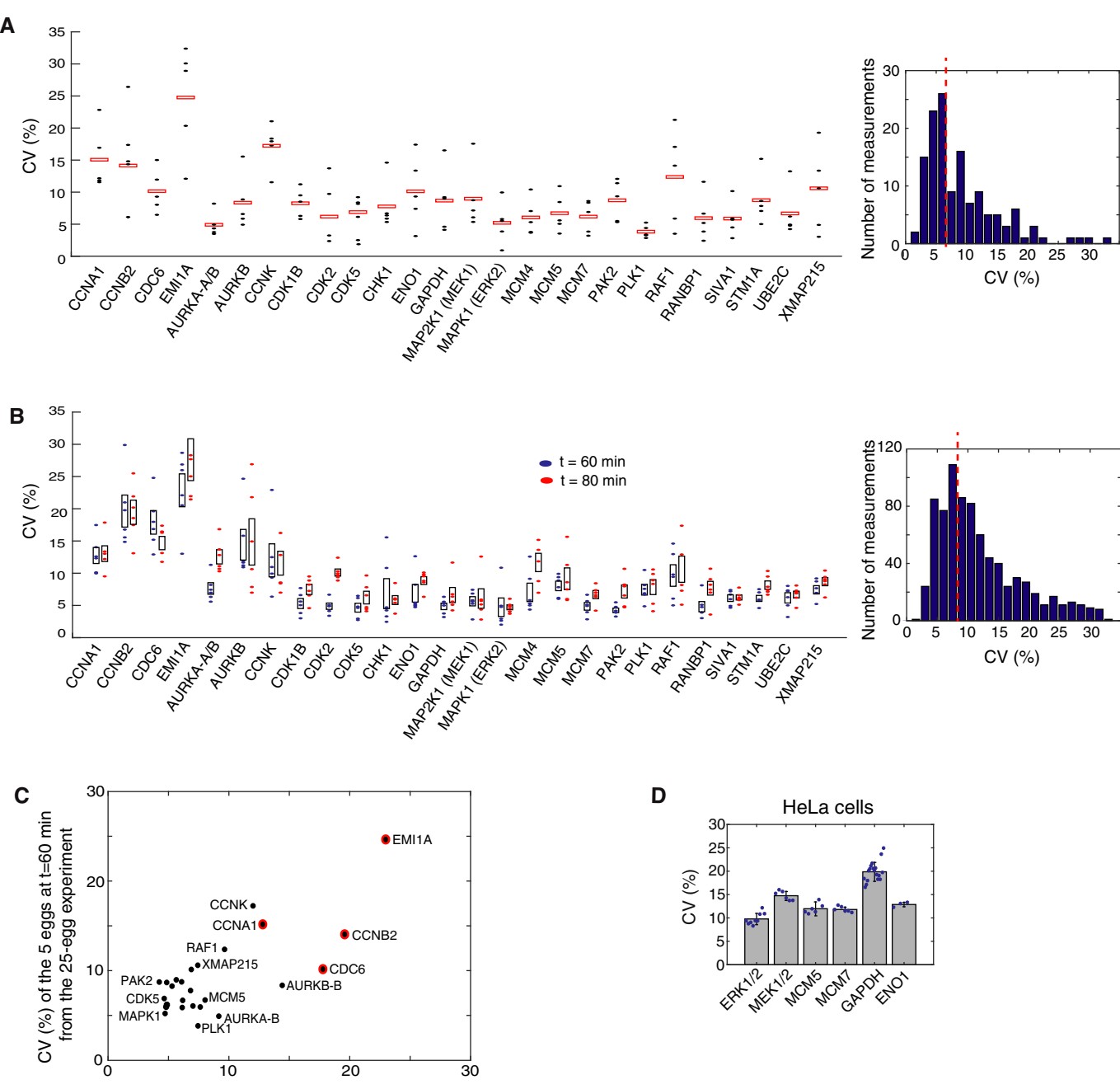

**Figure 3.  Single-cell variation in relative protein abundance is typically 5–10% in *Xenopus* eggs.**

A   Variation analysis of the relative abundance data from Fig 2C. Each point represents the coefficient of variation (CV) of the relative abundance of a protein between five individual eggs in a batch collected at the same timepoint. Red boxes mark the mean CV of 5 batches, each batch collected either at 0, 20, 40, 60, and 80 minutes after cell-cycle activation. (Right) The histogram of the measured CVs for all 26 proteins at five timepoints shows that CVs typically range from 5 to 10% with a mean CV of 7%.

B   Variation analysis of a second independent set of 120 eggs. Sixty individual eggs were collected at 60 (blue) and at 80 (red) minutes after the addition of calcium ionophore. The 60 eggs at each timepoint were divided into six batches of 10 eggs analyzed sequentially on the mass spectrometer to minimize technical variation. The CV of the relative abundance of each protein between 10 individual eggs in a batch was calculated and plotted as filled blue and red ovals. The black boxes mark the 25th to 75th percentile of the six batch-calculated CVs for each protein at either the 60- or 80-min timepoint. (Right) The histogram of the 312 CV measurements (6 CVs of 26 proteins at 2 timepoints) shows the mean CV is 9%.

C   Control scatter plot shows that the CVs of the 26 measured proteins are similar between two independent experiments: the 25-egg experiment shown in (A) and the 60-egg, 60-min experiment shown in (B). Red circles indicate proteins that have both high CV and change their abundance during the cell cycle.

D   CVs for a set of human homologs in HeLa cells. Immunocytochemistry was performed on cells plated in 96-well wells (representative images are shown in Fig EV5). Each blue dot represents the CV calculated from the ~5,000 cells in the respective well. Each barplot shows the mean CV of 3–12 wells. Error bars show standard deviation of the wells for that condition. Data shown are representative of three independent experiments.

co-regulation between MCM5 and MCM7 (Fig 4A and B), which is expected since they function as part of a stable MCM Helicase complex that can protect subunits from degradation in mammalian cells (McShane *et al*, 2016). Nevertheless, we were surprised to also find significant covariation between MEK (MAP2K1) and ERK (MAPK1) (Fig 4A and B) because such covariance adds extra noise to the signaling pathway and would not be beneficial for accurate analog signal transmission. As further validation of the statistical significant of the covariance, the *P*-values for MCM5/MCM7 and the MEK/ERK covariation remained significant, even after adjustment for multiple comparison testing by using Benjamini-Hochberg corrections (Table EV4).

To determine whether the covariances we observed in *Xenopus laevis* eggs are conserved in human cells, we carried out single-cell immmunohistochemistry measurements. As shown in Fig 4C, we found a strong covariance between MCM5 and MCM7. siRNA-mediated depletion experiments confirmed that MCM5 and MCM7 likely co-stabilize each other as both levels are reduced upon knockdown of either MCM5 or MCM7 in HeLa cells (Fig EV6). While control experiments showed weak covariation between MCM5 and the control protein GAPDH, we once again found a significant covariation between MEK and ERK, similar to the covariance we had observed in *Xenopus laevis* eggs (Fig 4C). This co-regulation is likely due to shared upstream expression regulation, or indirect feedbacks, as siRNA-mediated depletion of MEK and ERK showed opposing effects on ERK and MEK expression, respectively (Fig EV6). The unexpected covariation between MEK and ERK in both *Xenopus laevis* eggs and human cells made us consider whether it might be beneficial for a cell to have components of the same pathway covary, possibly in the context of binary cell activation that is often associated with MEK and ERK signaling pathways.

### Model analysis demonstrates that expression variation improves control of how many cells in a population make a binary cell-fate decision

As mentioned in the Introduction, previous studies showed that noise in signaling can be beneficial by widening the range of input stimuli that controls the percentage of cells in a population that are activated or not (Ahrends *et al*, 2014; Suderman *et al*, 2017). We were therefore interested to understand whether and how variation and covariation of the expression of pathway components could be main sources of noise for the control of binary cell activation. We first focused on variation and carried out simulations to understand the effect of variation of pathway components on binary signaling at the population level. As shown in the schematic in Fig 5A, we used the model introduced in Fig 1A but now assumed a last regulatory step whereby a cell with a $y_5^*$ value above 10 would trigger a switch into an active state while a cell with an output value $y_5^*$ below the threshold of 10 would remain inactive. This last step is denoted as B* versus B, reflecting the active and inactive binary output state, respectively. The results discussed here are largely independent of the value of the threshold (see Materials and Methods).

We used this binary model to determine the percentage of cells in a population that will switch into the active state for different fold-increases of input stimuli and different levels of expression variation. As shown by the black circles in Fig 5B, if there is no

expression variation of pathway components, all cells will reach the threshold and abruptly switch from the inactive to active state within a very narrow stimulus window. As the expression variation of pathway components increases and the cells become more different from each other, the percentage of cells in a population that switch from the inactive to active state can be controlled over a wider range of input stimuli. Increasing the CV of pathway components to 40% results in a close-to-linear relationship in the five-step model between percent of cells activated and relative input stimulus amplitude.

This widening of the input stimulus control window can be quantified by fitting the fractional activation data with an apparent Hill coefficient (aHC) that measures how well the population-level output can be controlled by the input. The fitted Hill coefficients for systems with different amounts of protein expression variation are shown in the bar plot in Fig 5C. A system with a smaller aHC can be more accurately controlled over a wider range of input levels which would be desirable in physiological settings where external hormone input stimuli may not be precise themselves. Another consideration to take into account is that physiological responses to hormone stimulation can typically be elicited over a 10-fold or greater range of relative hormone stimulus increases (R) (Atgiè *et al*, 1997; Katakam *et al*, 2001; Kimura *et al*, 2007). For a five-step signaling pathway, accurately transmitting a 10-fold range of input stimuli means that there should be a nearly linear relationship between input stimulus and percent of activated cells (Fig 5B). Such a broad and nearly linear relationship requires that the signaling pathway has high overall variation (approximately 40%) which could originate from variation in expression of individual pathway components or from other sources of noise.

### Understanding the role of variation in MEK and ERK expression in regulating bimodal ERK activity

Since the MEK/ERK signaling pathway often controls binary single-cell decisions such as whether cells divide or differentiate (Seger & Krebs, 1995), we used MEK and ERK as examples of variable signaling components to evaluate the role of expression variation in population-level cell-fate decisions. We first validated experimentally that ERK signaling output was bimodal or at least variable for intermediate stimuli in the same population by using EGF stimulation of human MCF10A cells. Specifically, we generated an MCF10A cell line expressing a FRET sensor of ERK activity to measure ERK activation in live cells (Albeck *et al*, 2013; Aoki *et al*, 2013). The FRET intensity of this sensor, EKAR-EV, was shown previously to faithfully report pERK levels in MCF10A cells (Yang *et al*, 2017). We used EGF to activate the pathway, and after 60 min, cells were fixed and stained with antibodies to measure the abundances of MEK and ERK, so that the pathway response could be related back to the relative level of the two proteins. The MEK and ERK abundance values were normalized by an intracellular total protein stain following established protocols from (Kafri *et al*, 2013) in order to correct for cell volume and to obtain relative protein abundances. An EGF titration showed that there was indeed bimodal ERK activation and that intermediate stimuli doses could induce heterogeneous responses (Fig 6A). We quantified the ERK activity in each timecourse by calculating integrated ERK activity as the area under the curve after EGF stimulation. As

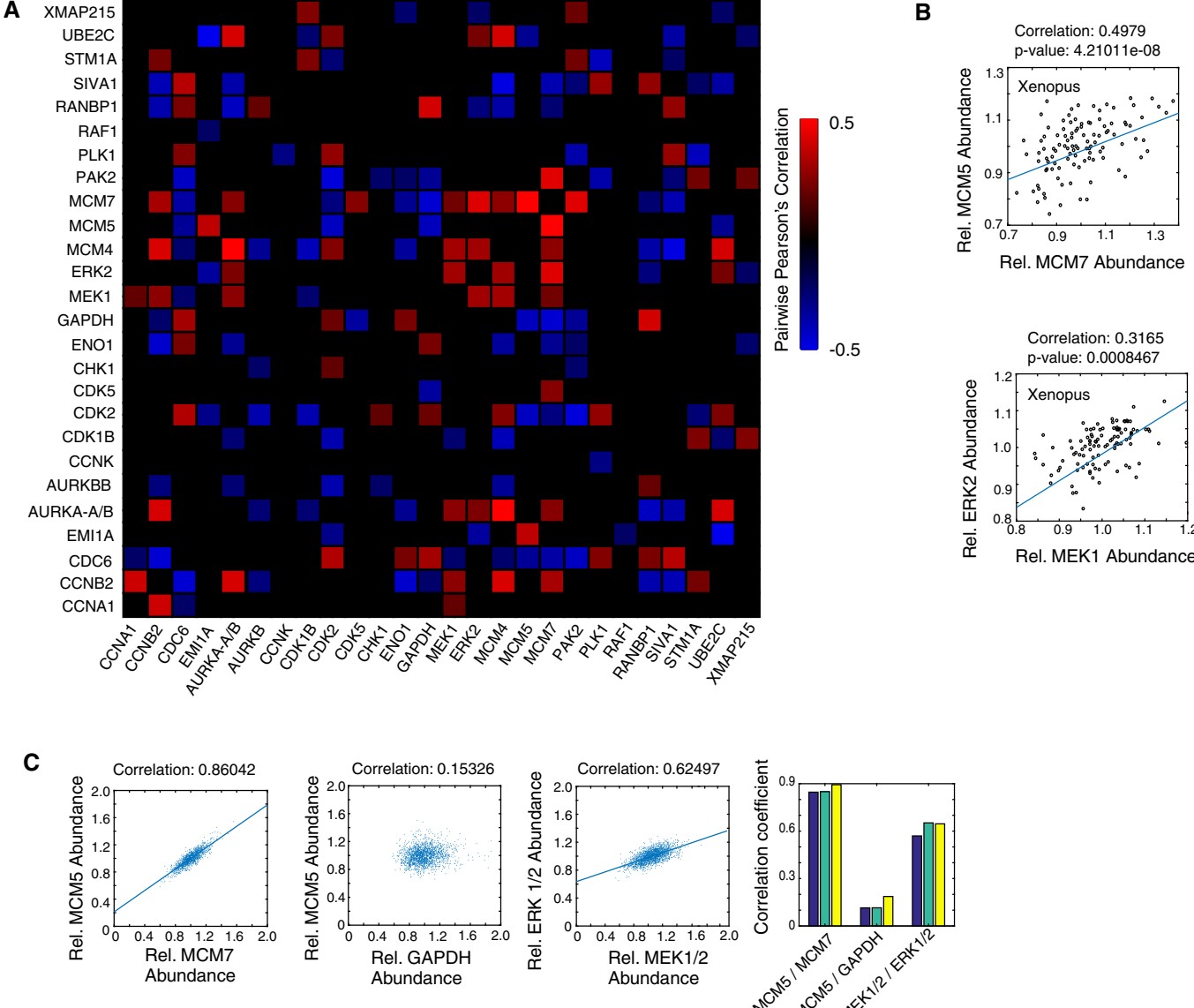

**Figure 4. MEK and ERK expression covary in *Xenopus* eggs and cultured human cells.**

A  Heatmap of Pearson's correlation values between the respective proteins in *Xenopus* eggs. Twenty-six relative protein abundances were correlated pairwise in 120 single eggs. Only correlations with a *P*-value less than 0.05 are shown. *P*-values were adjusted for multiple comparison testing using Benjamini-Hochberg corrections (Table EV4).

B  Two examples of pairwise correlations are shown between MCM5 and MCM7 and between MEK and ERK in *Xenopus* eggs.

C  Pairwise correlation analysis in HeLa cells, using MCM5 versus MCM7 as a positive control and MCM5 versus GAPDH as an uncorrelated control. Correlations between MEK and ERK concentrations are shown. Each scatter plot shows values from ~15,000 cells. The bar graphs on the right show correlation coefficients for three separate wells, containing ~5,000 cells each, for the same three correlation pairs.

shown in Fig 6A, the integrated ERK activity values showed two peaks, allowing us to define cells as active or inactive using the indicated threshold (dotted vertical black line). From the histograms in Fig 6A, it is apparent that the fraction of activated cells in the cell population increases as the EGF concentration increases. This relationship is more directly plotted in Fig 6B. Thus, in these human MCF10A cells, there is a wide range of input stimuli over which the fraction of the cell population that is activated can be controlled.

We next determined whether natural variation in MEK and ERK abundances indeed matters in determining whether individual cells have active ERK or not since it is conceivable that the level of active ERK is controlled by other factors such as variable numbers of receptors or variations in phosphatase activity. If expression variation matters for controlling activated ERK levels, the single-cell expression of MEK and ERK should on average be higher in cells with high ERK signaling compared to cells with low ERK signaling when analyzed in the same population of cells for the same

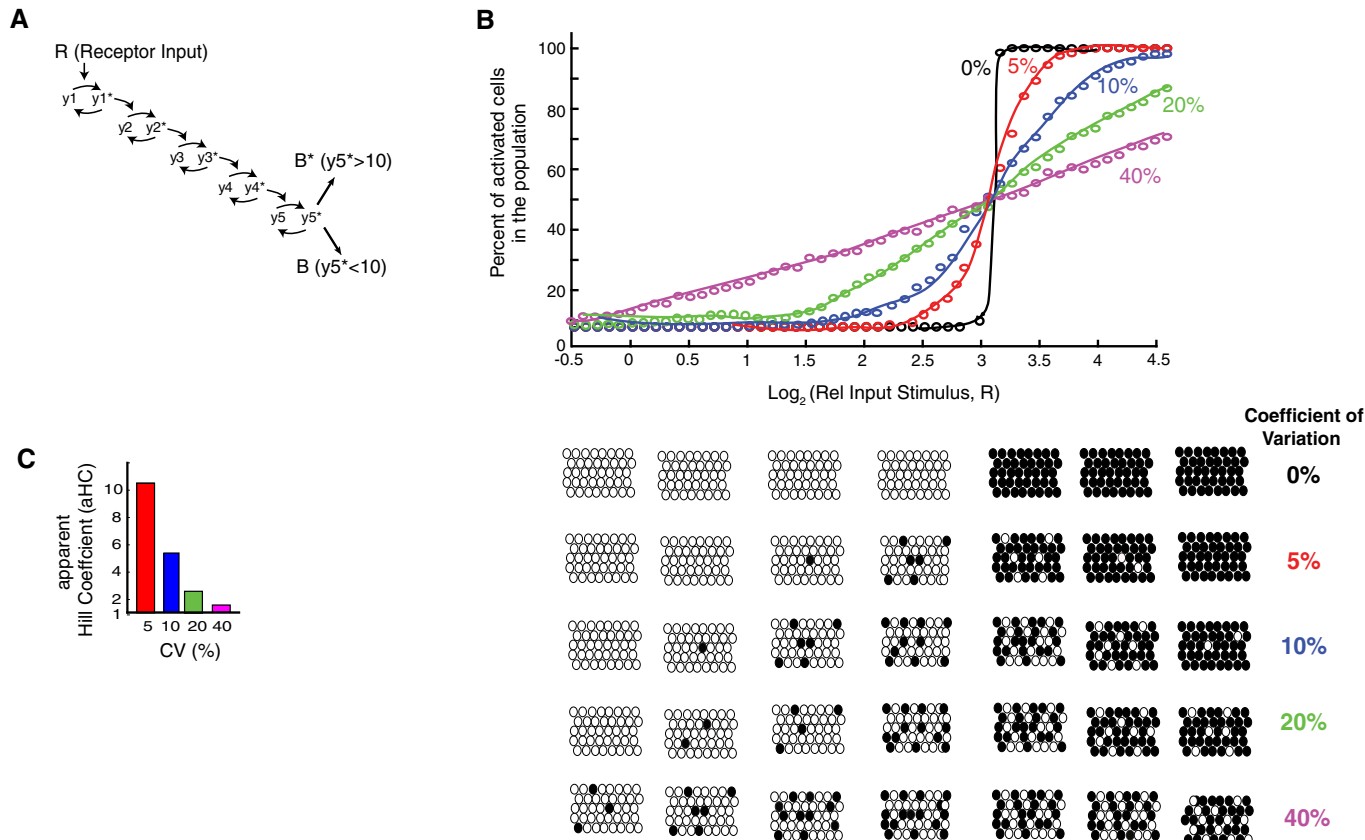

**Figure 5. Using a general five-step model to understand the effect of variation on controlling the fraction of cells in the population that respond to input stimulus.**

A   A binary output step was added to the model from Fig 1A. A threshold of 10 was used in each simulation to determine whether a cell was activated or not ($y_5^* > 10$).

B   Plot of how increasing the CVs in expression of the pathway components in this binary model from 0 to 40% increases the range over which changes in the input stimuli can change the fraction of cells in the population which trigger the binary switch and become activated.

C   Hill coefficients were fit to the data in (B) to quantify the steepness in the curves. The steepness is an inverse measure of how wide the input range is that controls the output.

intermediate input stimulus. Indeed, when we compared relative MEK and ERK abundances in cells with active or inactive ERK activity, we confirmed that activated cells have on average higher MEK and ERK concentrations, and inactive cells have on average lower MEK and ERK concentrations (Fig 6C). These results argue that natural single-cell variation in the concentrations of MEK and ERK does matter in determining whether or not a cell will be activated.

The five-step model in Fig 5 conceptually showed how expression variation can broaden the range over which input stimuli can control binary cell fates. We next used an established model of the MAPK pathway to better understand whether and how natural variation in MEK and ERK expression contributes to the controllability of bimodal ERK signaling in a population over a broader range of input stimuli. The model has seven protein species: Ras, MEK, ERK, four phosphatases, and RasGTP as the input (Sturm *et al*, 2010), and we added random lognormal noise with 10% CV to each simulation. We tested the model over a range of RasGTP input doses as a proxy for receptor input. When 15% random variation in both MEK and ERK was added to the model, the output of the model, phosphorylated ERK (pERK), which reflects ERK activity, became variable between cells and was bimodal for intermediate concentrations of

EGF stimuli as shown by the timecourse traces in Fig 6D. Figure 6E better illustrates the effect of adding variation to the model. The red and blue curves in Fig 6E show the percentage of cells with activated ERK at different doses of receptor stimulation when either 3 or 20% random variation in MEK and ERK was added. Increasing the expression variation of MEK and ERK in each simulation results in a more linear relationship between input stimulus and percent of activated cells, thus allowing for improved controllability of the percentage of activated cells in the population over a wider range of stimuli. Such a wide range over which stimuli regulate the cell function is important given that receptor stimuli have significant noise at the level of local ligand concentrations and receptor abundance.

### Understanding the role of covariation in MEK and ERK expression in regulating bimodal ERK activity

Given the need for low Hill coefficients and a broadening of the relationship between input stimulus and percent of activated cells in order to optimally control population-level responses to physiological stimuli (Fig 5B and C), we next determined whether covariation could be another source of overall noise that could lower the Hill

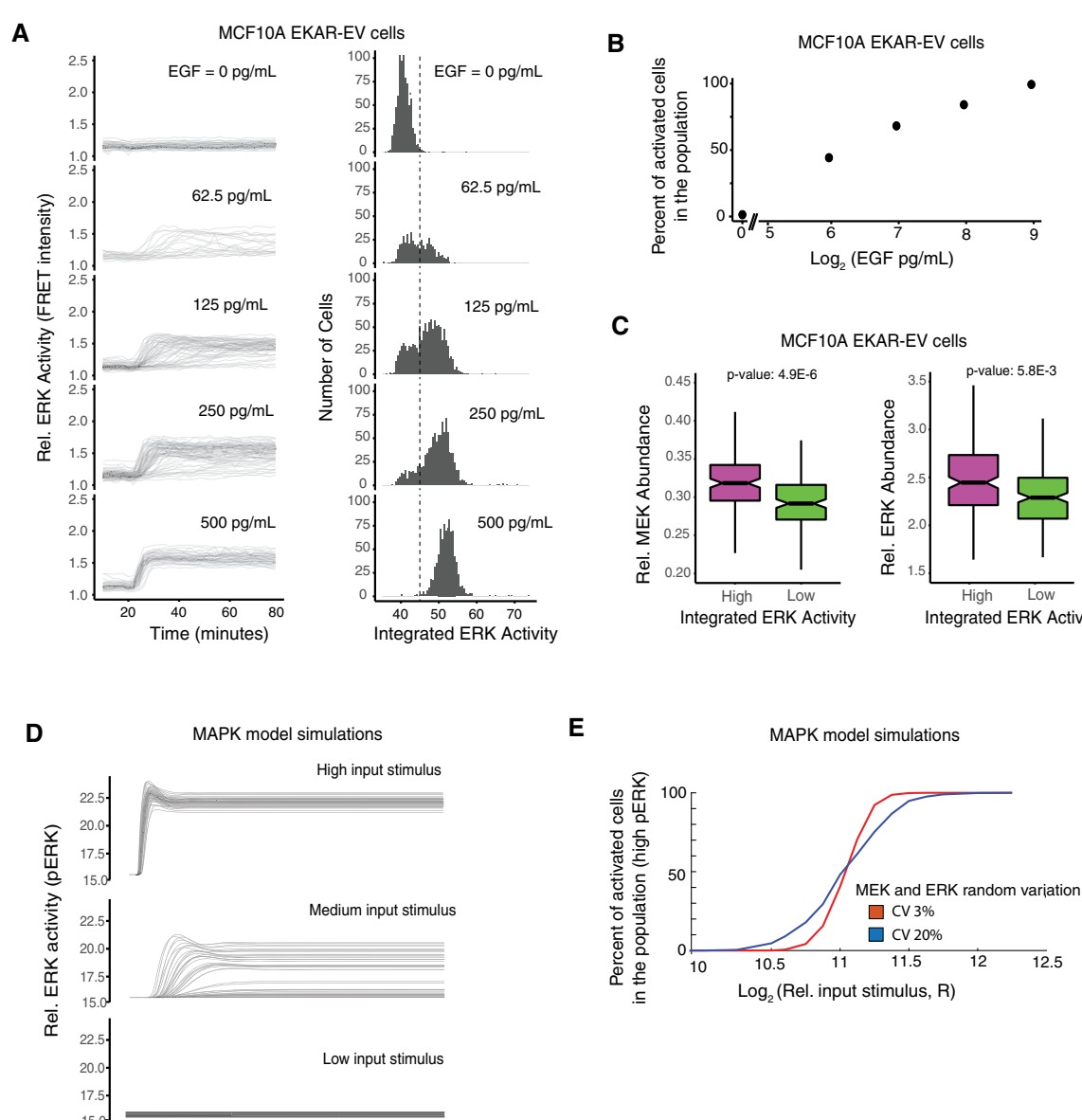

**Figure 6.  Live-cell imaging experiments and simulations using an established MEK/ERK signaling model show that variation between MEK and ERK expression widens the window over which input stimuli can control the fraction of cells that are activated in the population.**

A   MCF10A cells stably expressing the EKAR-EV FRET sensor were activated with varying concentrations of EGF after being serum starved for at least 48 h. Cells were imaged every 2 min throughout the timecourse. (left) The plots at EGF doses of 0, 62.5, 125, 250, and 500 pg/ml show FRET intensity timecourses from approximately 800, 520, 1,200, 1,000, and 900 individual cells, respectively. (right) Histograms show the corresponding integrated ERK activity of individual cells. Integrated ERK activity was calculated for each timecourse as the area under the curve after the addition of EGF. The dashed line shows the threshold used to distinguish cells with active versus inactive ERK.

B   Plot showing percentage of activated cells (cells to the right of the threshold plotted in (A)) in response to different EGF concentrations.

C   Box-and-whisker plots of MEK (left) and ERK (right) concentrations in cells with high (top 15%, magenta) or low (bottom 15%, green) integrated ERK activity in response to EGF stimulation. The high and low conditions represent 162 and 161 cells respectively, out of a total of 1,073 cells, stimulated with 3,000 pg/ml of EGF (MEK plots), and 198 and 197 cells respectively, out of a total of 1,316 cells stimulated with 125 pg/ml of EGF (ERK plots). In the box-and-whisker plots, the bold line in the center of the notch represents the median, the ends of the notched box represent the first and third quartiles, the length of the upper whisker shows the largest point no more than 1.5 times the inter-quartile range (IQR or length of the box), the lower whisker represents the smallest point no more than 1.5 times the inter-quartile range, and the notches represent 1.58 * IQR/sqrt(n), which approximates the 95% confidence interval of the median. The non-overlapping notches between the high and low populations, as well as the low *P*-values, indicate that the differences between the two populations are statistically significant.

D   Timecourse output from an established MEK/ERK model (Sturm *et al*, 2010) in response to high, medium, and low concentrations of input (RasGTP) stimulus shows that the output for intermediate stimuli is bimodal with mainly either high pERK or low pERK cells separated by a threshold pERK intensity of approximately 17. Random lognormal noise with 15% CV was applied to MEK and ERK and 10% CV to the input stimulus (RasGTP).

E   Model simulations resulting from applying random lognormal noise with different CVs to MEK and ERK. In all cases, random lognormal noise with 10% CV was applied to the input stimulus (RasGTP).

     

coefficient and improve controllability. Such an increase in overall noise is needed as a system with 10% expression variation may not generate sufficient signaling noise for accurate population control of binary signaling responses. We had shown in Fig 4C that MEK and ERK covary with each other in human HeLa cells. We now also confirmed that MEK and ERK covary with each other in the human MCF10A cells used for the FRET pERK activity measurements (correlation coefficient of 0.7; Fig 7A). Measurements of covariation between MCM5 and MCM7 and lack of covariation between MCM5 and GAPDH are shown as controls.

Next to understand the effect of covariation in a general multistep signaling pathway, we added covariation to the five-step model from Fig 5. As shown in the model output in Fig 7B, when covariation is added to all components in a regulatory system that has 10% variation of pathway components, the controllability of population-level binary responses is significantly improved by reducing the relationship between input stimulus and percentage of cells activated. This improvement in controllability is demonstrated by the Hill coefficient decreasing from over five if there is no covariation to down to 2.3 if covariation is added. Thus, our five-step model demonstrates that a system with high covariation of signaling components enables population-level regulation of binary outputs over a broader range of signaling inputs.

We next tested how strong the contribution is if only a pair of pathway components covaries by using the MAPK/ERK signaling model from Fig 6 and assuming that only MEK and ERK covary with each other. We compared how the fraction of activated cells in a population changes if MEK and ERK expression noise was random or covaried. We were cognizant that more than two pathway components may be co-dependent in typical signaling systems. As shown in Fig 7C (top plot), when there is covariation of a single pair of components, there was a small but significant broadening of the relationship between the stimulus intensity and the percentage of cells in the active state that seems to be particularly significant if cells have to control the activation of small fractions of cells. When considering cell differentiation as an example of this binary signaling response, a control of only 1% of the precursor population differentiates is critical physiologically since several tissues are believed to differentiate < 1% of precursor cells at any given time (Spalding *et al*, 2008; Ahrends *et al*, 2014). The significance of the contribution of covariation in this low percent range of cell activation can be seen by using a log scale for the *y*-axis in Fig 7C (bottom panel) and testing for the effect of applying noise to the input signal R. As shown in the panel, if one wants to keep 1% of the cells in the population activated (marked by the dotted horizontal line), a 10% difference in the input signal (represented by a black arrow) would result in less error in the number of activated cells (1.4-fold versus 2.3-fold accuracy in the percent of activated cells) when comparing a system with or without covariation in a single pair of pathway elements. Thus, a system with covarying components would be significantly more accurate in this physiologically relevant regime where low percentages of activated cells need to be maintained.

The model output in Fig 7D also confirms our experimental results from Fig 6C that cells in the population with high ERK activity have on average higher MEK and ERK levels compared to cells that have low ERK activity, arguing that the concentrations of MEK and ERK are limiting in the model and thus matter in determining whether or not a cell will be activated. Together, the plots in Fig 7C and D show that covariation of even one pair of pathway components—MEK and ERK in this case—significantly widens the range of input stimuli over which cell-fate decisions can be controlled at the population level. Covariation of more pathway components would further widen the stimulus range and further improve controllability. This result on the importance of expression variation and covariation is particularly important if organisms need to control the activation of small fractions of cells in a population such as to enable low rates of cell differentiation (Ahrends *et al*, 2014) or apoptosis (Spencer *et al*, 2009) in tissues.

## Constraints on accurate control of analog and binary signaling by expression variation and covariation

We used our experimentally measured low CV values for relative protein abundances, together with our finding that covariation can further improve the controllability of binary signaling outputs, to explore the respective ranges of variation and covariation where single-cell and population-level signaling can be effectively controlled. As depicted in Fig 8A, we employed a modification of the model from Fig 1A to directly compare analog and binary signaling outcomes by assuming that the same pathway drives in one case an analog single-cell output (A*) and, in the second case, binary cell activation if the output $y_5^*$ reaches higher than a threshold of 10 (B*). We use fold-Input Detection Limits (fIDLs), as defined in Fig 1C and D, to quantify accurate analog single-cell signaling and aHC, as defined in Fig 5C, to quantify accurate controllability of population-level binary signaling. As discussed in Fig EV1, the fIDL parameter is a measure of analog signaling accuracy that is inversely related to mutual information but is less dependent on the dynamic range of the output, and the Hill coefficient is an inverse measure of the input range over which the population-level output can be controlled. The equations used to calculate the fIDL and aHC are shown at the top of Fig 8B and C (see Materials and Methods for derivation).

As shown in Fig 8B and C, single-cell analog or population-level binary outputs can be optimally controlled if the fIDL or aHC, respectively, are small and close to 1. The conflicting constraint between the control of single-cell analog and population-level binary signaling by expression variation can be seen clearly by combining the two graphs in Fig 8B and C into a single competition curve (Fig 8D). Increasing the variation in the concentration of pathway components moves cells along this curve from optimal conditions for analog single-cell signaling (CV of 5%, right bottom) toward optimal conditions for controlling binary population-level signaling (CV of 40%, left top) with the curve staying far away from the origin at the left bottom where analog and binary signaling would both be accurate. Thus, the same signaling system with a CV of 5% that has optimal analog single-cell accuracy loses its ability to accurately control binary population-level outputs. Similarly, a system with a CV of 40% that is optimal for controlling binary population-level outputs loses its ability to accurately control analog single-cell signaling. Thus, cells cannot have a shared pathway that controls accurate analog single-cell signaling outputs and also accurately controls binary population-level signaling outputs.

As shown in Fig 8C and D, as well as in Fig 5B, a CV of 40% or greater would be optimal for controlling population-level signaling

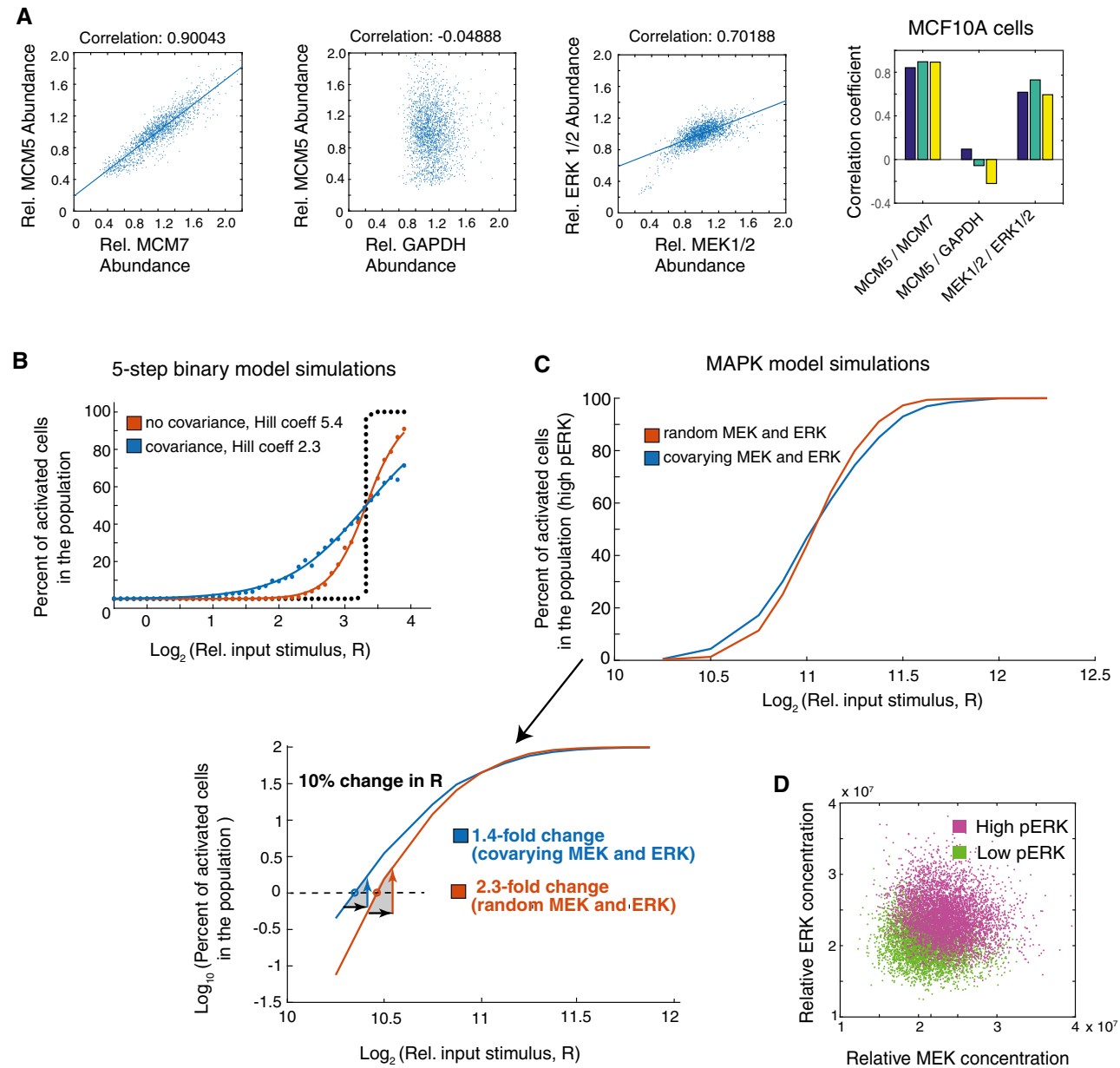

**Figure 7. Single-cell imaging experiments and model simulations show that covariation between MEK and ERK expression facilitates control of bimodal ERK activation.**

A   Immunohistochemistry experiments in MCF10A cells and pairwise correlation analysis show covariance of MEK and ERK. MCM5 versus MCM7 is used as a positive control and MCM5 versus GAPDH as an uncorrelated control. Each scatter plot shows values from ~15,000 cells. The bar graphs on the right shows correlation coefficients for 3 separate wells of each correlation pair, containing ~5,000 cells each.

B   Using the 5-step binary model from Fig 5A to now look at the effect of covariance in the pathway. The same type of plot as in Fig 5B is shown to compare the output of the binary model if the pathway components vary randomly or covary with each other. The population response when uncorrelated CVs of 10% were applied to the pathway components is shown in red. The blue curve shows the population response when covariation was added to the model. To obtain a maximal effect, the CVs of 10% were applied to all positive and all negative regulators, respectively, such that the positive regulators covaried together and the negative regulators covaried together. Covariation in the pathway broadens the range by which input stimuli can regulate the percent of activated cells, as shown by the decrease in the apparent Hill coefficient from 5.4 to 2.3 and less steep sigmoidal response.

C   Using an established MAPK model (Sturm *et al*, 2010) to compare the effect of covarying MEK and ERK concentrations. The red curve show the results of simulations in which random lognormal noise with 15% CV was applied independently to the MEK and ERK concentrations. The blue curve shows the results of simulations in which MEK and ERK concentrations were made to covary by applying the same 15% CV lognormal noise term to both MEK and ERK in each simulation. In all cases, lognormal noise with 10% CV was applied to the input stimulus (RasGTP). The shallower slope of the blue curve show that the percent of activated cells can be regulated over a wider range of input stimuli if there is covariance between MEK and ERK.

D   Output of simulations using same MAPK model as in (C). Scatter plot shows output of simulations (cells) colored by whether they had high (magenta) or low (green) ERK activity at the end of the timecourse. Cells shown were stimulated with input doses between $2^{10.5}$ to $2^{12}$, a range which results in both active and inactive cells in the population as shown in (C).

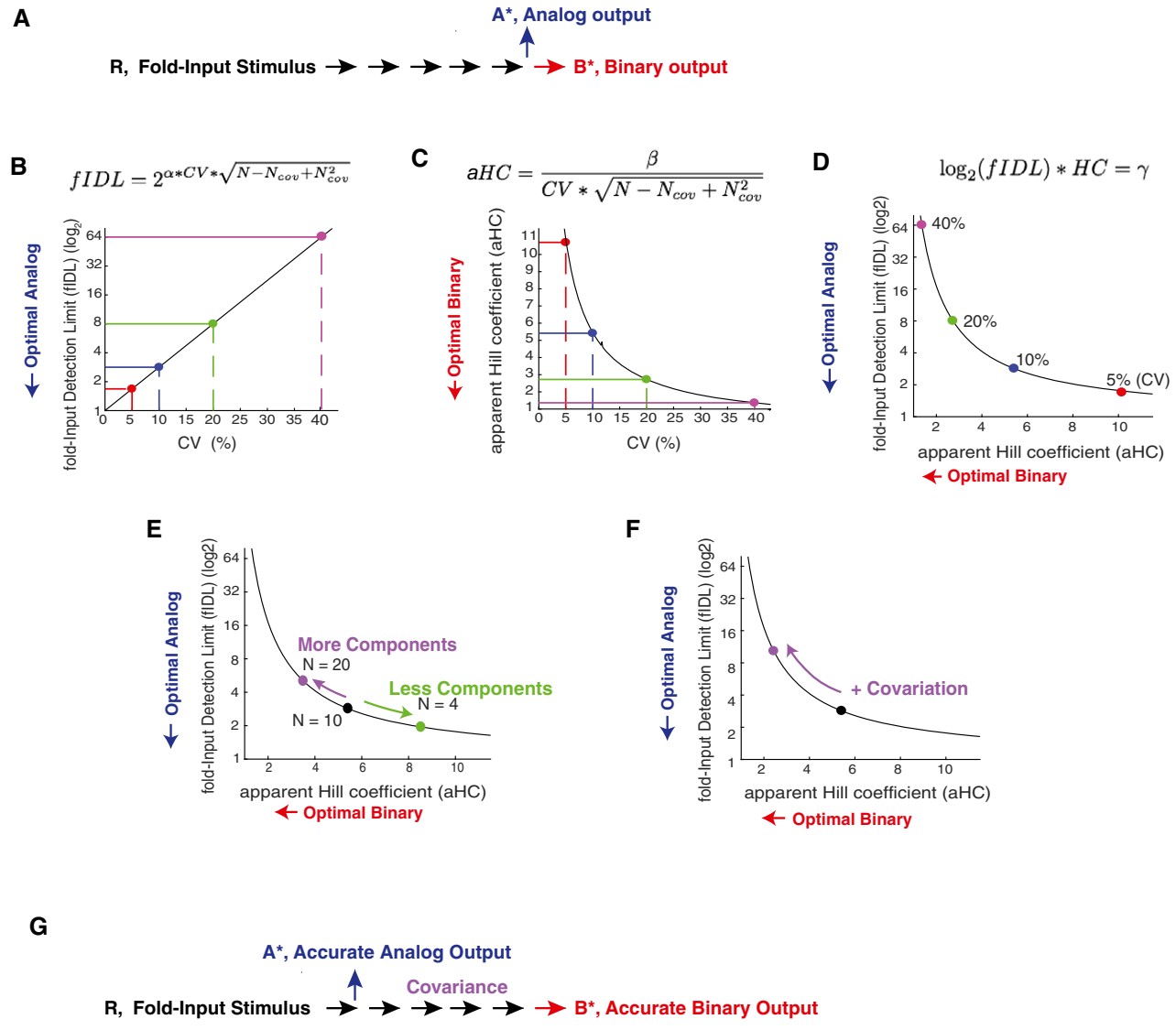

**Figure 8.  Competing demands on variation and covariation in the control of analog single-cell versus binary population-level signaling outputs.**

A       Schematic of a signaling pathway that splits into an analog or binary output.
B–D     Quantification of the competing constraints on expression variation for accurate control of single-cell analog versus population-level binary signaling outputs.
        Plots showing the development of a metric that quantitatively relates expression variation, analog single-cell signaling accuracy, and binary signaling
        accuracy. (B) Relationship between expression variation and fold-Input Detection Limit (fIDL), highlighting the physiological range of 5, 10, and 20% expression
        variation. (C) Relationship between expression variation and apparent Hill Coefficient (aHC). 40% expression variation enables accurate control of population-level
        binary outputs. (D) Integration of both relationships into a single co-dependency curve relating optimal analog single-cell and binary population-level signaling.
        Terms in equations: CV: expression variation; N: Total number of pathway components; $N_{cov}$: Number of covariant pathway components; α = 3.3; β = 1.4; γ = 6.3.
E       Increasing or decreasing the number of regulators in a pathway increases or decreases the overall noise in the pathway, respectively, and thus can be used as a
        way to more accurately control either binary population-level or analog single-cell functions, respectively.
F       Covariation between pathway components such as MEK and ERK is an effective means to increases overall noise in the pathway and thereby improve the
        controllability of binary population-level signaling responses while reducing single-cell analog signaling accuracy. A system with covariation can accurately control
        binary population level signaling without needing 40% expression variation which is likely not common.
G       The analysis of (B–F) suggests that the same pathway components can only be shared between analog and binary signaling systems if the analog pathway
        branches off early after receptor stimulation. Covariance in a branch of a signaling pathway is an indication that the output is regulated by a binary output at the
        population level.

outputs. However, our study and previous work by others suggests that such high CVs of protein concentrations are not common (Sigal *et al*, 2006; Gaudet *et al*, 2012), indicating that cells must use other mechanisms to generate the necessary high signaling noise to accurately control the fraction of activated cells for population-level

binary outputs. We considered that changes in the number of pathway components as well as the covariance of pathway components are strategies to alter the overall signaling output noise. We used the fIDL versus aHC co-dependency curve to determine how changes in pathway component numbers control analog or binary

signaling (Fig 8E). While our analysis so far assumed 10 regulatory elements, fewer or higher numbers of signaling steps are common in signaling systems. Notably, changing the number of signaling steps improves one signaling mode at the cost of the other. Fewer signaling steps move the system toward improved analog single-cell signal transmission and more signaling steps toward improved control of population-level binary outputs. To illustrate the effect of increasing or decreasing signaling steps with examples: since many signaling systems are complex with likely 20 or more regulators (Sturm *et al*, 2010; Gaudet *et al*, 2012), such complex systems must necessarily be mediating population-level signaling responses. In contrast, the visual signal transduction pathway in retinal cone cells, which transduces light intensity inputs proportionally into electrical outputs, has only a few main regulatory components (Arshavsky *et al*, 2002) which benefits the control of analog single-cell signaling responses.

Our modeling and experimental data in Figs 5 and 7 showed that a potent strategy to increase noise, without adding expression variation to individual components, is based on positive covariation between pathway components. Covariation can increase accurate binary signal transmission as we show in the case of the MEK/ERK signaling pathway. Indeed, Fig 8F shows that adding covariation moves cells away from a state where they can accurately perform analog single-cell signaling toward a state where they can accurately control the percentage of activated cells at the population level. These results suggest that covariation is a useful strategy to improve the control of population-level binary cell functions without that the expression variation or number of pathway components themselves have to be increased. We also note that covariation can in some cases increase rather than decrease, analog single-cell accuracy if directly opposing enzymes (e.g., a kinase and a phosphatase) covary with each other (Feinerman *et al*, 2008). Together, our analysis shows that cells have a versatile set of internal tools to control whether a signaling pathway can accurately control single-cell analog or population-level binary signaling by changing either the expression variation of individual components, the number of pathway components, or the covariation in expression between components. Furthermore, if pathways share components, these model calculations argue that analog signals have to minimize component numbers by branching out early in a pathway, while binary population-level signal responses would optimally be transmitted through more pathway steps and with pathway components covarying with each other (Fig 8G).

# Discussion

### Variation in signaling protein expression between individual cells is lower than expected

Variation in mRNA and protein expression between individual cells is believed to be a main limitation for cells to accurately transduce receptor inputs to control analog functional outputs. In particular, studies in model organisms and cultured mammalian cells suggest that the main sources of noise are likely the small number of mRNA molecules, which is common for signaling proteins, and the frequently observed bursting behavior in gene expression which can further increase the variation in the number of mRNA molecules

present in a cell at a given time. Together with the observed variation in the expression of fluorescently conjugated signaling proteins and the observed variation in antibody staining of signaling proteins, it has often been assumed that the CV of signaling proteins between cells in the same population must be quite high, with numbers of about 25% CV being frequently used.

Our study investigated signaling protein variation by measuring variation of a small set of signaling proteins in *Xenopus* eggs and a subset of proteins also in mammalian cells. Specifically, we measured variation by normalizing the expression of individual proteins by the total protein mass. Using this strategy, we found CV values for proteins between individual *Xenopus* eggs of 5–10% and between human cells in the 10–15% range, much lower than expected. We used in both cases total protein mass of a cell for normalization since cell volume is believed to scale closely with cell protein mass. Variation of the concentration is in most cases an optimal measure of variation, as the relevant parameter for the activity of a signaling protein is its cytosolic concentration, or the abundance of a particular signaling protein in a cell divided by the volume of the cell. The low CV values of 5–15% that we measured would make it possible for sensory, hormonal, or other analog signal transduction systems to accurately transduce information about the amplitude of an input to gradually control the output, as we demonstrated using a minimal model of a typical signaling pathway with five steps. As shown in Fig 1A–D, a five-step system with low variation in pathway component expression can with high accuracy distinguish a threefold increase in an input stimulus from a onefold increase, while a system with 25% variation can only distinguish a much larger fold-increase.

### Competing roles of expression variation in enabling accurate analog single-cell signaling and controlling population-level binary signaling

A main interest of our study was to better understand the competing requirements of analog signaling systems, that need to accurately control a gradual output response of a single cell, from binary signaling systems that need to control the percentage of cells in a population that trigger a particular cell-fate transition. In the analog single-cell case, low noise is optimal, while in the latter binary population-level case, high noise is optimal. In the binary case, the critical role of increased variation in the expression of signaling proteins is to broaden the response behavior in a population of cells so that the fraction of cells that trigger a cell-fate switch can be controlled over a broader range of input signals. If there is no noise, all cells would trigger the cell-fate change at precisely the same amplitude of an input stimulus. Given that input signals can also be noisy, one can argue that optimal binary systems should be able to control whether 10 or 50% of cells in a population are activated over a range of input stimuli of about 5, which would allow for approximately linear control of the fraction of activated cells by changes in the amplitude of typical receptor inputs. Our model calculations showed that such a system requires much higher variation in the expression of signaling proteins in the 40% range compared to the low variation required for optimal signaling for analog systems.

These competing needs for high versus low variation for different types of signaling raised the question whether cells use alternative mechanisms to increase overall signaling noise in a system in order to still allow cells to keep the variation of individual signaling

proteins relatively low if these same components are also needed in other situations for binary signaling. We show that having high numbers of signaling components involved in a binary decision is a powerful strategy to generate more noise as more components make a cumulative noise contribution that increases noise. We further showed in model calculations that covariation between signaling proteins in the same pathway reduces analog signal transmission accuracy but also found that covariation can both increase the overall noise/variation of the functional output. These considerations led us to measure protein covariation in *Xenopus* eggs and human cells, and we observed a significant covariation between MEK and ERK expression in both systems. Model calculations of the ERK pathway, together with measurements of ERK activity and ERK and MEK expression levels, showed that the increased variation due to covariation increases the range over which input EGF signals can control binary ERK activity output. Of note, the contribution from a single pair of covarying signaling proteins is relatively small, and strong effects resulting from covariation require multiple signaling proteins covarying with each other. Future studies with high quality antibodies for multiple pathway components will be needed to more generally test this covariation hypothesis in different signaling systems.

### Expression variation, covariation, and number of pathway components define accuracy of analog versus binary signaling systems

Another goal of our study was to develop a general formalism to better understand how variation and covariation of signaling protein expression and also the number of pathway can be modulated to optimize different types of signal transmission in cells. These factors impede the accuracy of single cell signal transmission while improving the controllability of the population-level regulation of binary cell activation. Our analysis in Fig 8 shows a clear competition arguing that cells should have relatively low variation of signaling proteins in the 10% range if they need to reuse these same pathway components also for population-level control of binary cell fates. This implied that other strategies are needed to increase the overall noise of the signaling pathway for the control of binary decisions. We identified that having large numbers of pathway components and having covariation between pathway components are two such strategies to increase the overall noise and to allow for population level control of cell fates over broad ranges of input stimuli. We developed a simple model that shows how many pathway components are needed and how much covariation can maximally contribute to the control of binary cell fates.

In conclusion, our study employed sensitive single-cell mass spectrometry and single-cell immunofluorescence analysis to reveal a low variation in relative protein abundances with CV values in the 5–15% range, suggesting that expression variation is not prohibitively high for analog signal transmission in single cells as was often assumed in previous studies. However, such low levels of variation make it difficult for signaling pathways to control population-level binary signaling outputs over broad ranges of input stimuli. We show that covariance of signaling components and increased numbers of pathway components can be effective mechanisms to increase overall signaling output noise and thereby allow for optimal control of binary cell-fate switches at the population level even if the variation of individual signaling components is low.

# Materials and Methods

### Reagents and tools table

| Reagent/Resource | Reference or source | Identifier or catalog number |
|---|---|---|
| **Experimental models** | | |
| MCF10A | ATCC | CRL-10317 |
| HeLa | ATCC | CCL-2 |
| *Xenopus laevis* | NASCO | |
| **Recombinant DNA** | | |
| pPBbsr2-EKAR-NLS | Komatsu *et al* (2011) | |
| **Antibodies** | | |
| Rabbit monoclonal antibody [E460] to ERK2 | Abcam | ab32081 |
| MEK1/2 (L38C12) Mouse mAb | Cell Signaling | 4694S |
| Rabbit anti-MCM5 | Abcam | ab17967 |
| Mouse anti-MCM7 | Abcam | ab2360 |
| Rabbit anti-ENO1 (Enolase) | Abcam | ab155102 |
| Anti-GAPDH antibody [6C5] mouse monoclonal | Abcam | ab8245 |
| Anti-GAPDH Polyclonal Goat anti-human, mouse, rat | Thermo Fisher | PA19046 |
| Alexa Fluor 647 NHS Ester | Thermo Fisher | A20106 |
| Donkey anti-Goat IgG (H+L) cross-adsorbed secondary antibody, Alexa Fluor 647 | Invitrogen | A21447 |
| Donkey anti-Rabbit IgG (H+L) cross-adsorbed secondary antibody, Alexa Fluor 488 | Invitrogen | A21206 |

Continued

| Reagent/Resource | Reference or source | Identifier or catalog number |
| --- | --- | --- |
| Donkey anti-Mouse IgG (H+L) cross-adsorbed secondary antibody, Alexa Fluor 594 | Invitrogen | A21203 |
| Donkey anti-Mouse IgG (H+L) cross-adsorbed secondary antibody, Alexa Fluor 647 | Invitrogen | A31571 |
| **Other** | | |
| siGENOME GAPD Control siRNA (Human) | Dharmacon | D-001140-01-05 |
| SMARTpool: siGENOME MAPK1 siRNA (ERK2) | Dharmacon | M-003555-04-0005 |
| SMARTpool: siGENOME MAP2K1 siRNA (MEK1) | Dharmacon | M-003571-01-0005 |
| SMARTpool: SiGenome MCM7 siRNA | Dharmacon | M-003278-02-0005 |
| SMARTpool: siGenome MCM5 siRNA (human) | Dharmacon | M-003276-02-0005 |
| SMARTpool: siGENOME MAPK3 siRNA (ERK1) | Dharamcon | M-003592-03-0005 |
| SMARTpool: siGENOME MAP2K2 siRNA (MEK2) | Dharmacon | M-003573-03-0005 |
| FlexiTube GeneSolution pooled siRNA for GAPDH | Qiagen | GS2597 |
| FlexiTube pooled siRNA for MCM5 | Qiagen | GS4174 |
| FlexiTube pooled siRNA for MCM7 | Qiagen | GS4176 |
| FlexiTube pooled siRNA for MAP2K1 (MEK1) | Qiagen | GS5604 |
| FlexiTube pooled siRNA for MAPK1 (ERK2) | Qiagen | GS5594 |
| ON-TARGETplus Non-targeting Pool Control siRNA | Dharmacon | D-001810-10-05 |
| Allstars Negative Control siRNA | Qiagen | 1027281 |
| JPT SpikeTides peptides | JPT Peptides, Berlin, Germany | |
| Epidermal Growth Factor (EGF) | Sigma-Aldrich | E9644 |
| PureCol Type I Bovine Collagen Solution | Advanced BioMatrix | Cat #5005 |
| Pierce BCA Protein Assay Kit | Thermo Fisher | 23225 |

## Methods and Protocols

### Xenopus laevis *egg collection and activation*

*Xenopus* egg extracts were prepared based on modifications of a previous protocol (Tsai *et al*, 2014). All of the animal protocols used in this manuscript were approved by the Stanford University Administrative Panel on Laboratory Animal Care. To induce egg laying, female *Xenopus laevis* were injected with human chorionic gonadotropin injection the night before each experiment. To collect the eggs, the frogs were subjected to pelvic massage, and the eggs were collected in 1× Marc's Modified Ringer's (MMR) buffer (0.1 M NaCl, 2 mM KCl, 1 mM MgCl$_2$, 2 mM CaCl$_2$, 5 mM HEPES, pH 7.8). To remove the jelly coat from the eggs, they were placed in a solution of 2% cysteine in 1× MMR buffer for 4 min and gently agitated, after which they were washed four times with 1× MMR buffer. To activate the cell cycle, eggs were placed in a solution of 0.5 µg/ml of calcium ionophore A23187 (Sigma) and 1× MMR buffer for 3 min, after which they were washed four times with 1× MMR buffer. Single eggs were collected at their respective timepoints and placed into 600 µl tubes and snap frozen in liquid nitrogen before being stored at −80°C.

### SRM sample preparation

Single eggs were lysed mechanically by pipetting the egg in 100 µl of lysis buffer (100 mM NaCl, 25 mM Tris pH 8.2, Complete EDTA-free protease inhibitor cocktail (Sigma). The lysate was then placed in a 400 µl natural polyethylene microcentrifuge tube (E&K

Scientific #485050) and spun at 15,000 *g* in a right angle centrifuge (Beckman Microfuge E) at 4°C for 5 min. The lipid layer was removed by using a razor blade to cut the tube off just beneath it, and the cytoplasmic fraction was pipetted into a 1.5-ml protein LoBind tube (Fisher Scientific #13-698-794), being careful to leave the yolk behind. To precipitate the proteins from the cytoplasmic fraction, 1 ml of ice cold acetone was added to each sample and placed at −20°C overnight.

To collect precipitated proteins, the samples were centrifuged at 18,000 *g* for 20 min at 4°C. Acetone was decanted, and the protein pellets were resolubilized in 25 µl of 8 M urea. To fully solubilize the protein pellet, the samples were placed in a shaker for 1 h at room temperature. The samples were then diluted to 2 M urea with 50 mM ammonium bicarbonate to a 100 µl volume, after which protein concentration was measured in duplicate with a BCA assay by taking two 10 µl aliquots of each sample. The proteins in the remaining 80 µl of sample volume were reduced with 10 mM TCEP and incubated for 30 min at 37°C, then alkylated with 15 mM iodoacetamide and incubated in the dark at room temperature. Next, the samples were diluted to 1 M urea with 50 mM ammonium bicarbonate, and heavy peptides (JPT SpikeTides) were added based on BCA assay results. Trypsin (Promega #V5113) was then added at a ratio of 10 ng trypsin per 1ug protein (no < 500 ng was added to a sample). The trypsin digestion was carried out at 37°C for 12–16 h.

To stop the trypsin, formic acid (Fisher #A117-50) was added at a ratio of 3 µl per 100 µl of sample to bring the pH down to < 3.

Peptides were cleaned up using an Oasis HLB uElution plate (Waters), equilibrated, and washed with 0.04% trifluoroacetic acid in water, and eluted in 80% acetonitrile with 0.2% formic acid. All solutions used are HPLC grade. Samples were then lyophilized. To remove any variance produced by phosphorylated peptides, the samples were phosphatase-treated. Peptides were resolubilized in 50 μl of 1× NEBuffer 3 (no BSA), and calf intestinal alkaline phosphatase (NEB #M0290S) was added at a ratio 0.25 units per μg of peptide and incubated for 1 h at 37°C. The peptides were cleaned up again according to steps described above. Peptides were resolubilized in 2% acetonitrile and 0.1% formic acid before SRM analysis.

### SRM data acquisition

As detailed in previous publications (Abell et al, 2011; Ahrends et al, 2014), 2 μg of peptides was separated on an EASY-nLC Nano-HPLC system (Proxeon, Odense, Denmark) with a 200 × 0.075 mm diameter reverse-phase C18 capillary column (Maisch C18, 3 μm, 120 Å) and were subjected to a linear gradient from 8 to 40% acetonitrile over 70 min at a flow rate of 300 nl/min. Peptides were introduced into a TSQ Vantage triple quadrupole mass spectrometer (Thermo Fisher Scientific, Bremen, Germany) via a Proxeon nanospray ionization source. The transitions for the light (endogeneous) and heavy (SpikeTide) peptides were measured using scheduled SRM-MS and analyzed using Skyline version 3.5 (MacCoss Lab, University of Washington). Relative peptide quantifications were determined by rationing the peak area sums of the transitions of the corresponding light and heavy peptides. Only transitions common between the heavy and light peptides with relative areas that were consistent across all samples were included in the quantification. Lists of transitions used for the 25-egg measurements in Figs 2C and 3A and C and for the 120-egg measurements in Figs 3B and 4A are given in Tables EV2 and EV3, respectively.

### SRM data statistical analysis

To minimize sample processing differences, a maximum of 30 single eggs were prepped and analyzed at the same time by SRM mass spectrometry. While we normalized the amount of heavy reference peptides added to each egg extract to the measured single egg protein concentration, this leaves still a small measurement error between individual eggs. This is likely both a result of small errors in the measurement of protein concentration and small volume pipetting errors, causing small under- or overestimation of relative protein abundances in a sample. This small calibration error was in previous protocols corrected using a normalization factor measured as a median of a set of anchor protein peptides (Abell et al, 2011; Ludwig et al, 2012; Feng & Picotti, 2016). Here, we used the median of 22 normalized peptide intensities that minimally change during the cell cycle to derive a concentration correction factor for each egg (this factor was typically between 0.9 and 1.1). The lack of change in expression of these proteins during the cell cycle can be seen in Fig 1D. The correction we used makes the assumption that the 22 peptides are not overall co-regulated in the same direction, an assumption that is supported by both our SRM-MS and immunohistochemistry experiments (Fig 4). Specifically, we measured for a set of analyzed single eggs (e.g., 25 eggs in Fig 2A) the medians of the relative abundances for each of the 22 peptides across all eggs. To obtain a correction factor for each egg, we first normalized each peptide by the median of that particular peptide across all samples of interest (e.g., for the 25-egg analysis show in Fig 1D, each peptide value was first divided by the median of that peptide across all 25 samples). Then, we calculated the median of the 22 normalized peptide values for each egg. The resulting correction factor value was typically in the range of 0.9 to 1.1, and we divided all 26 relative protein abundances from that egg by this factor. The variation and covariation values shown in this paper use these corrected relative abundances.

### Cell culture

MCF10A cells (ATCC, CRL-10317) were cultured in a growth media consisting of DMEM/F12 (Invitrogen) supplemented with 5% horse serum, 20 ng/ml EGF, 10 μg/ml insulin, 0.5 ng/ml hydrocortisone, 100 ng/ml cholera toxin, 50 U/ml penicillin, and 50 μg/ml streptomycin. HeLa cells were cultured in DMEM (Invitrogen) plus 10% fetal bovine serum (FBS) and penicillin–streptomycin–glutamine (PSG).

### EKAR-EV-NLS stable cell line

pPBbsr2-EKAR-EV-NLS was described previously (Komatsu et al, 2011). To generate stable cell lines, the construct was co-transfected with the piggybac transposase vector into human MCF10A cells using polyethylenimine. Cells with stable integration of the vector were selected for using 10 μg/ml blasticidin (Invivogen).

### Immunofluorescence

Cells were fixed by adding paraformaldehyde to the cell media for 15 min (final concentration of paraformaldehyde in media was 4%). Cells were then washed three times in PBS before they were permeabilized by adding 0.2% triton X-100 for 20 min at 4°C before being washed again with PBS. To remove cell size effects, cells were then stained with Alexa 647 NHS Ester as a marker of total protein mass and surrogate for cell volume/thickness following protocols described in (Kafri et al, 2013). The Alexa 647 NHS Ester was added at a concentration of 0.04 μg/ml in PBS for 1 h. After washing again in PBS, a blocking buffer consisting of 10% FBS, 1% BSA, 0.1% triton X-100, and 0.01% NaN$_3$ in PBS was added, and the cells were incubated for 1 h at room temperature. Then, primary antibodies were added overnight at 4°C, followed by incubation with secondary antibodies for 1 h at room temperature. To obtain particular protein concentrations for each cell, the mean total cell intensities of the respective antibodies were ratioed over the mean total cell intensity of the Alexa 647 NHS Ester.

### siRNA transfection

siRNAs were used at a final concentration of 20 nM and are listed in the Reagents and tools table. MCF10A and Hela cells were reverse-transfected with siRNA using Lipofectamine RNAiMax according to the manufacturer's instructions. The cells were fixed 48 h after reverse transfection with siRNA.

### Image acquisition

For both fixed and live-cell imaging, cells were plated in 96-well, optically clear, polystyrene plates (Costar #3904). Approximately 10,000 HeLa cells or 5,000 MCF10A cells were plated per well. For MCF10A cells, the wells were first coated with collagen (Advanced

BioMatrix Cat #5005, PureCol Type I Bovine Collagen Solution) by placing 50 μl of collagen dissolved at a ratio of 1:100 in PBS in each well, incubating for 2–3 h at room temperature, and then rinsing three times with PBS. MCF10A cells were then plated into the wells in MCF10A growth media. For assays to determine EGF responses, the media were aspirated from the cells 24 h after plating and replaced with serum starvation media for 60 h (DMEM/F12, 0.3% BSA, 0.5 ng/ml hydrocortisone, 100 ng/ml cholera toxin, PSG). For imaging, the cells were placed into an extracellular buffer consisting of 5 mM KCl, 125 mM NaCl, 20 mM Hepes, 1.5 mM $MgCl_2$, 1.5 mM $CaCl_2$, and 10 mM glucose. Time-lapse imaging was performed initially in 75 μl of extracellular buffer per well to which an additional 75 μl of extracellular buffer containing 2× EGF doses was added to stimulate the cells. Cells were imaged in a humidified 37°C chamber at 5% $CO_2$. Images were taken every 2 min in the CFP and YFP channels using a fully automated widefield fluorescence microscope system (Intelligent Imaging Innovations, 3i), built around a Nikon Ti-E stand, equipped with Nikon 20×/0.75 N.A. objective, an epifluorescence light source (Xcite Exacte), and an sCMOS cameras (Hammamatsu Flash 4), enclosed by an environmental chamber (Haison), and controlled by SlideBook software (3i). Five non-overlapping images were taken per well.

### Image processing and analysis

#### Segmentation and tracking

Cell segmentation and tracking were performed using the "MACK-track" package for MATLAB available at http://github.com/brookstaylorjr/MACKtrack, and described in (Selimkhanov *et al*, 2014). In place of the first-step cellular identification using differential interference microscopy, the first pass whole-cell segmentation was performed here by thresholding the total protein stain image.

#### Signal measurement

Four-channel fluorescence images were taken with a 10× objective on a MicroXL microscope, and image analysis was performed using MATLAB analysis. Background subtraction was used in the Hoechst (to stain DNA and mask the nucleus), the two immunofluorescence, and the protein mass fluorescence channels. Signal intensities were corrected for non-uniformity but were still restricted to a central R = 350 pixel region of 2 × 2 binned images (1,080 × 1,080 pixels) of the image to minimize potential spatial non-uniformities in illumination and light collection toward the corners. The Hoechst stain was used to establish a nuclear mask and to select cells in the 2N G0/G1 state based on the integrated DNA stain. The Hoechst intensity levels used to define cells in the 2N state were selected by inspection of the Hoechst histograms. The live-cell FRET measurements of nuclear ERK activity were performed on a Nikon Ti2 controlled by 3i software (Intelligent Imaging, Denver, CO). The mean nuclear intensities of the FRET and CFP channels were ratioed for each cell to obtain the normalized FRET value at each timepoint. At the end of the timecourses, the cells were fixed and stained with either an ERK or MEK antibody, as well an Alexa 647 NHS Ester as an estimate of cell volume. To obtain ERK and MEK concentrations for each cell, the mean total cell intensities of the ERK and MEK antibodies were ratioed over the mean total cell intensity of the Alexa 647 NHS Ester. The final ERK and MEK concentrations for each cell were then matched to the corresponding FRET timecourse for that particular cell.

### Modeling

#### Figure 1A and B

The goal of these figure panels is to illustrate how different amounts of noise (cell-to-cell variation) would affect the output of a multi-step linear signaling system. We used MATLAB simulations to apply expression variation in the concentrations of pathway components in a five-step linear signaling pathway with a single input and output, representing a typical vertebrate signaling pathway. The model is not saturated and uses a single fold-input R to increase pathway activation linearly above the basal activity level. The last regulated signaling step y5 is shown as the analog output A*. We simulated protein expression variation of each of the 10 signaling pathway components using lognormal Monte Carlo noise simulations (each of the 10 system parameters was multiplied by randomly variable factors centered on 1). We followed the system over time using the ODE45 function until it reached equilibrium at $t = 15$.

Linear model:

$$\frac{dy_1^*}{dt} = \epsilon_1 * R * y_1 - \epsilon_2 * y_1^* \qquad (1)$$

$$\frac{dy_2^*}{dt} = \epsilon_3 * y_1^* * y_2 - \epsilon_4 * y_2^* \qquad (2)$$

$$\frac{dy_3^*}{dt} = \epsilon_5 * y_2^* * y_3 - \epsilon_6 * y_3^* \qquad (3)$$

$$\frac{dy_4^*}{dt} = \epsilon_7 * y_3^* * y_4 - \epsilon_8 * y_4^* \qquad (4)$$

$$\frac{dA^*}{dt} = \epsilon_9 * y_4^* * A - \epsilon_{10} * A^* \qquad (5)$$

R is the Receptor Input into the cell that activates $y_1$.

A* corresponds to $y_5^*$ and denotes the final output signal (i.e., final Signaling Response of the cell). Each signaling step acts linearly on the next intermediate step.

The model is not saturated. For each step, we assume that the active $y^*$ states are generated from a relatively larger constant pool of precursor cells. In other words, $y_i$, as well as A, denotes a pool of inactive precursors that is not significantly diminished during signal transmission ($y_i$ is approx. equal to $y_i$, total and A is approx. equal to Atotal) and is set equal to 1.

To introduce uncorrelated lognormal noise into the system:

$$\epsilon_i = e^{(randn*CV)} \qquad (6)$$

For $i = 1$–$10$, randn is a lognormally distributed random number and CV is the percent noise in the system, typically from 5 to 25%. We are introducing noise into the system as lognormal since we are assuming that the noise sources are multiplicative not additive in the system (i.e., work to change the enzyme rates) which is a reasonable assumption in biology.

For $|x| << 1$, $e^x \approx 1 + x$, which keeps the CV of the real distribution approximately the same as the CV of the lognormal distribution.

The ten lognormal stochastic values of a factorial parameter e(1-10) are calculated for each of typically 5,000 runs to generate the plots, e(i:10)=(exp(randn(10,1), Var) in MATLAB. Var is the percent

variation parameter that changes in different panels in the plots. Calculating a coefficient of variation (CV) of the resulting random parameter distribution returns the value Var.

### Figure 1C and D

This figure illustrates how the fold-Input Detection Limit (fIDL) is calculated for a particular noise level (CV) and Receptor Input. We assume there are N independent pathway components which increases the overall noise in the output by noise propagation to:

$$CV_{total} = CV * \sqrt{N} \tag{7}$$

The calculation of fIDL was done analytically using the inverse normal distribution function in MATLAB to determine the fraction of cells in a population that are in the desired tail region of the output probability distribution. The resulting value is half of the required signal output amplitude since both the unstimulated and stimulated distributions are symmetrical when they are plotted as a log scaled distribution. The factor 2 in the equation reflects that basal and stimulated output distributions are assumed to have the same noise (see Fig 1C, black and green distributions). When assuming 95% accuracy for distinguishing stimulated and unstimulated cells from the output signal, and assuming N independent components in the system, the resulting fold-Input Detection Limit is calculated as:

$$fIDL = exp^{(2*norminv(0.95)*CV*\sqrt{N})} = exp^{(\alpha*CV*\sqrt{N})}; \alpha \approx 3.3 \tag{8}$$

### Figure 1E

We also compared uncorrelated variation versus correlated variation (covariation) between signaling components in the pathway. In Fig 1E, we made the assumption that the five positive elements and the five negative elements in the model (possibly reflecting protein kinases versus protein phosphatases) each have a correlated variation. We compare this to the case were all variations are independent of each other as we also do in all other figure panels. This correlated variation leads to an increase of the overall variation of the signaling response of a cell.

The model is the same as shown in Fig 1A and B and Equations (1–5), except that lognormal noise is added into the system as follows:

$$\epsilon_1 = \epsilon_3 = \epsilon_5 = \epsilon_7 = \epsilon_9 \tag{9}$$

$$\epsilon_2 = \epsilon_4 = \epsilon_6 = \epsilon_8 = \epsilon_{10} \tag{10}$$

### Figure 5A–D

The goal of these figures is to show how variation and covariation of proteins in a pathway contribute to the control of binary, population-based signaling responses. We simulated binary pathways in Fig 5 by using the model from in Fig 1 and adding an assumption that cells trigger a binary switch when the output $y_5^*$ exceeds a threshold level of 10, and cells do not trigger the switch when the final output remains below this threshold level. The threshold of 10 was chosen arbitrarily (the results presented are largely independent of the value of this threshold). We used increasing fold-stimuli

strength R and analyzed in the simulations the increasing fraction of cells that triggers the switch. Figure 5B plots the percent of "activated cells" (i.e., the fraction of cells out of all simulations that resulted in an output level > 10) versus the strength of input stimulus R. The solid lines show best fit using a Hill equation. The resulting best fit Hill coefficients are shown in the bar plot in Fig 5C.

Similar to the calculation of a fIDL in Fig 1C and D, we noticed in our simulations that one can describe these Hill plots by an "apparent Hill coefficient" (aHC) that is over a broad range of thresholds largely independent of the threshold value used as long as the threshold is larger than the total noise of the system:

$$aHC = \frac{\beta}{CV * \sqrt{N}}; \beta \approx 1.4 \tag{11}$$

### Figure 6A–C

The goal of this figure is to show how covariation of MEK/ERK abundance improves the sharpness of the binary population-level outcome. For numerical simulations, we used the ODE model of the ERK signaling network from (Sturm *et al*, 2010) with negative feedback intact. The model incorporates dynamics from RasGTP through Raf and MEK down to ERK phosphorylation. We used the input concentration of RasGTP as a proxy for extracellular EGF. The output was defined as doubly phosphorylated ERK (pERK), which serves as a proxy for ERK activity, as ERK activity is a monotonically increasing function with respect to pERK.

### Figure 8

The goal of this figure is to illustrate the conflicting effects that expression variation, covariation, and number of pathway components have on controlling analog single-cell and binary population-level signaling responses. The equations for fIDL and aHC were presented earlier (Equations 8 and 11). We are now also combining these two curves by multiplying the logarithm of fIDL with aHC to show their co-dependency. The shown combined error is derived from an error propagation analysis for correlated and uncorrelated variation of pathway components:

$$log2(fIDL) * aHC \approx 6.3 \tag{12}$$

### Figure EV1

Figure EV1 compares fIDL values to the log2 mutual information content of the same system (bits), adding different levels of saturation to the last term of the equation (as an example, we used instead of y(4) the term 10*y(4)/(y(4)+9) for the system that imposes a saturation of a factor of 10 to the output signal). For the mutual information calculations, 10,000 simulations were made with R values spread out using a random number generator in log2 units. Output A* (log2 units) were simulated, and the mutual information was derived from R and A* by using log2 in the MI equation and by using binning of 0.05 for R and A*.

### Data and software availability

Data and MATLAB analysis scripts used in this paper are available at https://github.com/Teruellab/Kovary_MSB_2018.

**Expanded View** for this article is available online.

## Acknowledgements

This work was supported by National Institutes of Health RO1-DK101743, RO1-DK106241, and P50-GM107615 (to M.N.T.), Stanford BioX Seed Grant funding (to M.N.T.), F32-DK114981 (to B.T.), T32-NIH T2HG00044 and 1F31DK112570-01A1 (to M.L.Z.). We thank Xianrui Cheng and Graham Anderson for help with *Xenopus* protocols and James Ferrell, Tobias Meyer, and members of the Teruel Lab for discussions and critical reading of the manuscript.

## Author contributions

KMK, MLZ, BT, and MNT conceived experiments. KMK and BT performed experiments. KMK, BT, and MNT carried out model simulations. KMK, MLZ, BT, and MNT analyzed data. KMK and MNT wrote the paper with input from all authors.

## Conflict of interest

The authors declare that they have no conflict of interest.

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
