## [Review Process File · Molecular Systems Biology]

Expression variation and covariation impair analog and enable binary signaling control

Kyle M. Kovary, Brooks Taylor, Michael L. Zhao and Mary N. Teruel

Review timeline:

Submission date:	17 September 2017
Editorial Decision:	31 October 2017
Revision received:	16 December 2017
Editorial Decision:	7 February 2018
Revision received:	26 March 2018
Accepted:	5 April 2018

Editor: Thomas Lemberger

Transaction Report:

1st Editorial Decision

31 October 2017

Thank you again for submitting your work to Molecular Systems Biology. We have now heard back from the three referees who agreed to evaluate your manuscript. As you will see from the reports below, the referees find the topic of your study of potential interest. They raise, however, substantial concerns on your work, which, I am afraid to say, preclude its publication in its present form.

The reviewers find that the question addressed by the study is of potential interest, in particular given the experiments on mammalian cells. They remain however unconvinced that some of the major conclusions are sufficiently supported by the data. They thus raise a series of important criticisms on the analysis and interpretation of the data (eg Fig 4) and the conclusiveness of some experiments (eg siRNA).

If you feel you can satisfactorily deal with these points and those listed by the referees, you may wish to submit a revised version of your manuscript. Please attach a covering letter giving details of the way in which you have handled each of the points raised by the referees. A revised manuscript will be once again subject to review and you probably understand that we can give you no guarantee at this stage that the eventual outcome will be favorable.

On a more editorial level, the imaging data, quantification data and the model used should be provided to ensure of the reproducibility of the reported results.

- We would thus also encourage you to include the source data for figure panels that show essential data, so that readers can download these data directly from the figure. Source data files are associated to individual panels of main figures. *Numerical data* should be provided as individual .xls files (including a tab describing the data) or csv or tab-delimited text files. *For 'blots' or

microscopy*, uncropped images should be submitted. For *network visualization*, Cytoscape session files, if available, can be supplied. The files should be labeled as "Source Data for Figure 1A" etc. Source Data for Expanded View and Appendix figures should be uploaded as a single ZIP file containing all the Source Data for Expanded View and Appendix content. (Additional information on source data is available in the "Guide for Authors" section at <http://msb.embopress.org/authorguide#sourcedata>).

- As you may have noticed, we recently replaced Supplementary Information by Expanded View (EV, see examples in <http://msb.embopress.org/content/11/6/812>). In this format, a limited number of Supplementary Figures (max 5) can be integrated in the article as EV figures that are interactively collapsible/expandable and will be typeset by the publisher. In this case, the figures should be cited as 'Figure EV1, Figure EV2' etc... in the text and their respective legends should be added to the main text after the legends of regular figures. The illustrations should be provided as separate files.

- For the figures that you do NOT wish to display as Expanded View figures items, they should be bundled together with their legends in a 'traditional' supplementary PDF, now called the *Appendix*. Appendix should start with a short Table of Content and the figures should be named and referred to in the main text as: "Appendix Figure S1, Appendix Figure S2" etc. See detailed instructions regarding expanded view here: <http://msb.embopress.org/authorguide#expandedview>.

- Additional Tables/Datasets should be labeled and referred to as Table (or Dataset) EV1 etc. Table/Dataset legends can be provided in a separate tab in case of .xls files. Alternatively, you can upload a .zip file containing the Table/Dataset file and a separate README .txt file with the legend/description.

When you resubmit your manuscript, please download our CHECKLIST (http://embopress.org/sites/default/files/Resources/EP_Author_Checklist_Master.xlsx) and include the completed form in your submission. *Please note* that the Author Checklist will be published alongside the paper as part of the transparent process <http://msb.embopress.org/authorguide#transparentprocess>.

REVIEWER REPORTS

Reviewer #1:

In their manuscript „expression variation and covariation impair analog and enable binary signaling control", Kovary and colleagues show that cell-to-cell variation of proteins/kinase concentrations is smaller than previously reported - some protein concentrations co-vary - and discuss potential implications for quantitative or binary signalling by making model simulations. The authors focus on MAPK signalling, and first use quantitative proteomics on *Xenopus* oocytes, where they show that the single cell coefficient of variation is about 10% for ERK and MEK, as well as many other proteins. They show that ERK and MEK are correlated in *Xenopus*, as well as in human cell lines, and use model simulations that show that at 10% of CV, analogue signalling may still be possible, but due to correlations it is not. They then make simulations that show that when signalling is binary, on a population level signalling may be graded (with more and more cells responding), and perform life-cell imaging with subsequent quantification of MEK and ERK to validate that "on" cells have correlated higher expression levels. Some simulations then show in which case analogue and binary signalling can work.

Overall, I find the *Xenopus* quantification convincing, but have problems following the human cell line work, in particular the key point of the paper (Figure 4), as detailed below. Also, I think the presentation of the paper is suboptimal, and it would benefit from a thorough rewrite, and obscures some of the problems of the analysis.

Major points:

1) In figure 4, the variance and co-variance of ERK/MEK (in panel 4J) is much higher than the variance that has been estimated in Figure 3 for these proteins. The authors rightly normalise their staining data with an intracellular total protein stain to correct for cell volume in Figure 3, but use a

different normalisation strategy in Figure 4 (median intensity). This suggests that there is a confounding variable leading to correlated ERK/MEK levels, and this confounding variable might actually be the reason for responders/non-responders. This could e.g. be cell size, thickness, state in cell cycle or response to the ligand (cell grows after it received the stimulus). To rule this out the same normalisation as in Fig. 3 needs to be applied. Alternatively, manipulation of the expression levels of ERK and MEK could be used to see if it influences the response.

2) It should be established that the reporter actually works in all cells in reporting pERK, which is the output of the model (e.g. by co-staining with pERK)

3) While in *Xenopus* the topology of the signalling network is possibly nicely depicted by the scheme in Figure 1 (linear cascade), signalling is far more complex and includes feedbacks in mammals. The authors do use the Sturm model, but the variance in the simulation of ERK and MEK seems to be very high (compared to their 10% that they estimate in Figure 3). This is not surprising, as Sturm et al. and Fritsche-Günther et al. show that the feedback makes the pathway extremely robust against inhibitors and variations in protein concentration. Thus, Figure 4E/H and 4F/I actually show that when strong feedback is involved, one needs far higher (co-variation) in protein levels to get somewhat binary signalling.

4) Many of the plots related to the simulations are uninterpretable. E.g. in Fig. 4E-G it is completely impossible to estimate how many cells respond (are red), how many don't (blue) and how many are in the middle (i.e. are not following their "binary paradigm").

Minor:

1) The storyline currently builds up a contradiction between analogue and binary that is not supported by literature. For *Xenopus*, the colleagues at the same institution (Ferrell) have shown very convincingly in a series of landmark papers that ERK signalling is binary (switch-like) at the single cell level and analogue at the population level. These should be cited and the story shouldn't start suggesting that signalling in oocytes is analogue!

2) There are no x-axis tick labels in Fig. 3D.

3) The standard-deviation or variance estimator is unbiased, i.e. its expectation value converges to the "correct" value. However, the distribution of the estimator is heavily skewed, therefore the median of several standard-deviations is biased towards lower values. Use mean instead!

4) many cross-references to panels in figure 1 are wrong

5) technical terms like variation, co-variation and covariance are not correctly used in many places (e.g. CV is not variation but the coefficient of variation)

6) in legend of fig. 3C each blue dot is not the mean of 5000 cells, but is the CV calculated from 5000 cells

7) figure captions are often more an interpretation of the data/simulations, where instead they should explain what is shown in the figure

8) The whole story is built under the assumption that ERK signalling functions on the population level. It is unclear in which contexts MAPK signalling should work at the cell population level, and why this should be graded.

Reviewer #2:

In this manuscript, Kovary and coworkers investigate the variation in protein abundance of signaling proteins. By performing SRM mass spectrometry measurements in single *Xenopus* eggs, they show that protein abundances typically varies by around 7% between individual cells. Additionally, some proteins from the same pathway, including MCM5 and MCM7 as well as MEK1 and ERK2, show significant co-variance. This finding was verified in HeLa cells using immunocytochemistry. By mathematical modeling using a simple five-step kinase model as well as a previously established RAF/MEK/ERK model (Sturm et al., *Sci Signal*, 2010), they show how covariation influences signaling outputs: low protein variance enables an accurate analog (gradual) signaling response, while high covariation allows to control the fraction of cells showing a binary (all-or-nothing) response on the population level.

The authors address an important topic in signal transduction and the single cell mass spectrometry experiments have been performed very carefully. The article is nicely written and the concepts are well explained. However, some issues require clarification.

Major comments:

1. Figure 1F: What is the technical error in determining the protein abundance of these proteins? To test this, several mass spectrometry experiments with identical lysates could be performed. On page 17, the authors conclude that "the relative abundance of components of signaling pathways still vary by at least 5%". It should be ascertained that this lower limit is really biological and not technical.
2. Figure 2B: show separate boxes for time point 60 and time point 80 min. Figure 2C: To show that the two independent experiments show the same results, the same time points should be compared. Therefore, it would be better to show the CV of time point 60 min only on the y-axis.
3. Page 8, second paragraph, "we added covariation to the model": It is unclear how this covariation was added. Do the protein abundances perfectly correlate? In the results of Figure 3B, correlation was 0.53 and 0.38. More importantly, for the results in Figure 2D, the authors assume that the kinases OR the phosphatases covary together. As it is correctly pointed out on page 17, if the kinases covary with the phosphatases, the conclusions are completely different. Therefore, the impact of kinases AND phosphatases covarying together should also be modeled.
4. Figure 3C-D: Immunocytochemistry measurements have been performed, but only the analyzed data is shown. At least some examples of the acquired images should be shown.
5. Page 12, second paragraph, "seven species in the MAPK model ... whose coefficient of variation were chosen to match the experimental data": Where is the experimental data for the seven species? Only the distributions of MEK and ERK were measured.
6. Figure 4B: Why did the authors select such an unrealistically low ($r=0.04$) and high ($r=0.99$) correlation between MEK and ERK? In the experiments shown in Figure 3B and 3D, correlations of 0.38 and 0.62 have been measured for MEK and ERK, while for the uncorrelated proteins GAPDH and MCM5 a correlation of 0.15 was measured.
7. Page 13, first paragraph, "the correlated distribution had increased numbers of low responding cells and high-responding cells": In Figure 4D (paired), a peak at the highest possible integrated ERK activity can be seen. This presumably corresponds to an unrealistic complete ERK activation as seen in the plot with the highest ppERK level in Figure 5F. Such unrealistic simulations should be avoided. The same holds true for the high-frequency oscillations seen in Figure 5E and 5F.
8. Figure 4E-G: The authors claim that when MEK and ERK are correlated, the abundance of MEK correlates with the high responding population. An alternative explanation would be that only the abundance of ERK is predictive of a low versus high response and, if MEK and ERK are correlated, the same holds true for MEK. Therefore, the conclusion on page 13, second paragraph, "with covariation having a key role in controlling which cells are activated and which ones are not" is too strong.
9. Page 14, first paragraph, "the two proteins proved equally able to separate the two subpopulations of cells". According to the box plot in Figure 4J, there is no significant difference between the two subpopulations.

Minor comments:

1. Abstract and Page 2: The unit of protein expression variation ($\pm 25\%$) should be added, presumably CV is meant.
2. Page 3, five lines from bottom, "to accurately population-level decisions": verb is missing.
3. Page 5, seven lines from bottom, "an fIDL stimulus of 2.8": 2.83 would be better to match the figure.
4. Page 7, second paragraph, "measurements in 25 individual *Xenopus laevis* eggs collected at 5 timepoints": please re-phrase to indicate that measurements have been performed at 5 time points with 5 eggs each.
5. Figure 1F: The number of dots of each timepoint is sometimes less than five. Are these missing values from the mass spectrometry experiments?
6. Figure 2A: The median should be removed from the scatter plot as there are only five data points. However, the total median (7%) could be added as a line in the histogram.
7. Figure 2B: Individual boxplots for $t = 60$ min and $t = 80$ min should be plotted. The median should be indicated with horizontal lines rather than with ovals.
8. Page 9, second paragraph, "covariation between MEK and ERK in cultured human cells": It should be mentioned that these are HeLa cells.

Reviewer #3:

Kovary et al. present an analysis of the effect of protein expression variation on signal transduction pathways. Beginning with a model analysis, they motivate their study with simulations showing that the oft-cited 25% CV for expression variance shown in earlier studies would make it impossible for a 5-step signaling cascade to accurately relay stimulus levels. They then turn to experimental measurements of expression variance, first using *Xenopus* oocytes as a base case, and show a very nice set of measurements, with a median CV of 7%, arguing that low expression variance enabling more accurate signaling is possible within a biological system. They also observe co-variation between MEK and ERK expression levels, a finding they extend to cultured mammalian cells. They note using simulations that such co-variation can in fact enhance variance in signaling output, which is important in the case where the binary responses must be controlled at the population level, and they then test this hypothesis with an elegant experiment in which variable responses to a low dose of EGF, measured by live-cell reporter, are correlated with MEK and ERK expression levels in the same cell after fixation, providing support for the idea that the expression co-variance plays a part in the variable signaling response. Finally, they bridge the concepts of low variance as an important factor for accurate signaling at the single cell level and high variance as necessary for control of the fraction of cells undergoing a binary fate decision using a mathematical analysis of their cascade model.

Overall I found this manuscript thought provoking. It presents a nice exploration and synthesis of several ideas about the role of expression variance in signaling and some interesting experimental data that are in general supportive of the conceptual ideas. The mass spec study in particular seems very carefully performed. However, there are several points that should be addressed in revision:

1) The data in EV3 raise some important questions about the ERK/MEK immunofluorescence experiments. In the quantification of siRNA knockdowns, only a 30-40% reduction is seen for either MEK or ERK by siRNA. When this is presented as a validation of the antibody staining, it leaves open the question of whether the remaining signal is non-specific antibody staining or just incomplete knockdown. It is also not very convincing to have only a single siRNA for each gene. The conclusion that knockdown of MEK affects ERK and vice versa is questionable given the relatively small effect sizes and the well-known off-target effects of siRNA. The important points here could be made much stronger if additional knockdowns giving more complete depletion could be shown. Also, why are MEK and ERK shown as bar graphs rather than histograms as for MCM5/7? If knockdown is not uniform across the population, that would certainly be relevant to the interpretation of the results.

2) The data in Fig. 4J are fairly impressive but, given their importance to the argument, need to be better controlled. It would be helpful to see one or more of the control stains (GAPDH, MCM5/7) to establish how an unrelated protein would behave in this experimental setup. It would also be helpful (perhaps as a supplement) to see simulations such as in Fig. 4H/I with different degrees of randomization of MEK and ERK to help assess where the experimental result lies in the continuum between the two extremes of unsorted vs. fully sorted.

3) The ordering and grouping of the results might be reconsidered. While the writing is clear enough that I could follow the logic throughout, the narrative does feel a bit winding as it jumps from variance as a theoretical limit on signaling accuracy, to showing low variance in the oocytes, to the need for co-variance at the population level. It seems likely that readers less familiar with the subject area would be confused by this. Contributing to this feeling is that the data in the first four figures are grouped a little oddly, with a new theme being introduced in the last few panels of each figure and then continued in more detail in the next figure.

4) Various typos: "do to" instead of "due to" on page 9; "know" instead of "known" on page 7; missing word in "to accurately population level decisions" on page 3.

Reviewer #1:

In their manuscript "Expression variation and covariation impair analog and enable binary signaling control", Kovary and colleagues show that cell-to-cell variation of proteins/kinase concentrations is smaller than previously reported - some protein concentrations co-vary - and discuss potential implications for quantitative or binary signaling by making model simulations. The authors focus on MAPK signaling, and first use quantitative proteomics on *Xenopus* oocytes, where they show that the single cell coefficient of variation is about 10% for ERK and MEK, as well as many other proteins. They show that ERK and MEK are correlated in *Xenopus*, as well as in human cell lines, and use model simulations that show that at 10% of CV, analogue signaling may still be possible, but due to correlations it is not. They then make simulations that show that when signaling is binary, on a population level signaling may be graded (with more and more cells responding), and perform life-cell imaging with subsequent quantification of MEK and ERK to validate that "on" cells have correlated higher expression levels. Some simulations then show in which case analogue and binary signaling can work.

Overall, I find the *Xenopus* quantification convincing, but have problems following the human cell line work, in particular the key point of the paper (Figure 4), as detailed below. Also, I think the presentation of the paper is suboptimal, and it would benefit from a thorough rewrite, and obscures some of the problems of the analysis.

We have reorganized the manuscript and rewritten key transition sections in respond to the reviewer's comments.

Major points:

1) In figure 4, the variance and co-variance of ERK/MEK (in panel 4J) is much higher than the variance that has been estimated in Figure 3 for these proteins. The authors rightly normalize their staining data with an intracellular total protein stain to correct for cell volume in Figure 3, but use a different normalization strategy in Figure 4 (median intensity). This suggests that there is a confounding variable leading to correlated ERK/MEK levels, and this confounding variable might actually be the reason for responders/non-responders. This could e.g. be cell size, thickness, state in cell cycle or response to the ligand (cell grows after it received the stimulus). To rule this out the same normalization as in Fig. 3 needs to be applied. Alternatively, manipulation of the expression levels of ERK and MEK could be used to see if it influences the response.

We have now normalized the data in Figure 4 (now Figure 7) by using the intracellular total protein stain in the same manner as we did for the data in the original Figure 3.

2) It should be established that the reporter actually works in cells in reporting pERK, which is the output of the model (e.g. by co-staining with pERK)

The FRET reporter was shown to faithfully report pERK levels in MCF10A cells by our departmental colleagues in Tobias Meyer's lab. To show that the FRET levels reported pERK levels, they measured FRET in the EKAR-EV cells and then co-stained with pERK antibody for a range of EGF doses. The linear relationship between nuclear pERK (measured with the pERK antibody) and ERK activity measured with the FRET reporter is shown in Supp. Fig. 2D in their recent Nature paper (Yang et al., 2017). We now stated in the manuscript on p. 15: "The FRET intensity of this sensor, EKAR-EV, was shown faithfully report pERK levels in MCF10A cells (Yang et al., 2017)."

3) While in *Xenopus* the topology of the signaling network is possibly nicely depicted by the scheme in

Figure 1 (linear cascade), signaling is far more complex and includes feedbacks in mammals. The authors do use the Sturm model, but the variance in the simulation of ERK and MEK seems to be very high (compared to their 10% that they estimate in Figure 3). This is not surprising, as Sturm et al. and Fritsche-Günther et al. show that the feedback makes the pathway extremely robust against inhibitors and variations in protein concentration. Thus, Figure 4E/H and 4F/I actually show that when strong feedback is involved, one needs far higher (co-variation) in protein levels to get somewhat binary signaling.

We apologize for the lack of clarity and changed the analysis. We made sure that concentration variations (variations of the respective parameters) are in the 10-15% range as observed experimentally (lognormal). Specifically, we have now made the simulations using 15% variation for MEK and ERK. We would like to note that even with the feedback there is apparent bimodality in the simulated ERK activity output especially when we covary ERK and MEK concentration.

4) Many of the plots related to the simulations are uninterpretable. E.g. in Fig. 4E-G it is completely impossible to estimate how many cells respond (are red), how many don't (blue) and how many are in the middle (i.e. are not following their "binary paradigm").

We agree that the figure was confusing, and we have now redone it to make it more clear (new Figure 6). We have also included histograms to be able to better show the number of cells in responding and non-responding categories.

Minor:

1) The storyline currently builds up a contradiction between analog and binary that is not supported by literature. For *Xenopus*, the colleagues at the same institution (Ferrell) have shown very convincingly in a series of landmark papers that ERK signaling is binary (switch-like) at the single cell level and analogue at the population level. These should be cited and the story shouldn't start suggesting that signaling in oocytes is analog!

Yes, we do talk to Jim Ferrell very often, especially since his office is right next to Mary Teruel's. Jim Ferrell told us that MAPK/ERK signaling is likely both binary and analog in *Xenopus* depending on the developmental stage. MAPK/ERK signaling is important in *Xenopus* for maturation - where it is binary - and then it shuts off. It then becomes important again in the *Xenopus* blastula/gastrulation stage (Curran and Grainger, 2000; LaBonne and Whitman, 1997) where it could be involved in analog signaling.

2) There are no x-axis tick labels in Fig. 3D.

Thank you for noticing this. We have fixed the x-axis tick labels.

3) The standard-deviation or variance estimator is unbiased, i.e. its expectation value converges to the "correct" value. However, the distribution of the estimator is heavily skewed, therefore the median of several standard-deviations is biased towards lower values. Use mean instead!

We have now used the mean instead of the median in all plots in Figure 3 where we are comparing coefficient of variations (CV's).

4) many cross-references to panels in figure 1 are wrong

We have corrected the cross-references to the panels in Figure 1 in the manuscript.

5) technical terms like variation, co-variation and covariance are not correctly used in many places (e.g. CV is not variation but the coefficient of variation)

We have corrected the use of the variation terms in the manuscript.

6) in legend of fig. 3C each blue dot is not the mean of 5000 cells, but is the CV calculated from 5000 cells?

Fig. 3C is now Fig. 3D, and yes, in this figure, each blue dot represents the CV calculated from the ~5000 cells in the respective well. We have now stated this in the figure legend.

7) figure captions are often more an interpretation of the data/simulations, where instead they should explain what is shown in the figure

Thank you for bringing this point up. We have now fixed the figure legends to omit interpretation of the data/simulations.

8) The whole story is built under the assumption that ERK signaling functions on the population level. It is unclear in which context MAPK signaling should work at the cell population level, and why this should be graded.

ERK has been shown to regulate cell-fate changes such as proliferation (Yang et al, 2017, Nature), differentiation, oncogene-induced senescence, and apoptosis in many tissues and organisms which often implies that at the cell population level, only a fraction of precursor cells would at any time undergo such a cell-fate change.

Reviewer #2:

In this manuscript, Kovary and coworkers investigate the variation in protein abundance of signaling proteins. By performing SRM mass spectrometry measurements in single *Xenopus* eggs, they show that protein abundances typically varies by around 7% between individual cells. Additionally, some proteins from the same pathway, including MCM5 and MCM7 as well as MEK1 and ERK2, show significant co-variance. This finding was verified in HeLa cells using immunocytochemistry. By mathematical modeling using a simple five-step kinase model as well as a previously established RAF/MEK/ERK model (Sturm et al., *Sci Signal*, 2010), they show how covariation influences signaling outputs: low protein variance enables an accurate analog (gradual) signaling response, while high covariation allows to control the fraction of cells showing a binary (all-or-nothing) response on the population level.

The authors address an important topic in signal transduction and the single cell mass spectrometry experiments have been performed very carefully. The article is nicely written and the concepts are well explained. However, some issues require clarification.

Major comments:

1. Figure 1F: What is the technical error in determining the protein abundance of these proteins? To test this, several mass spectrometry experiments with identical lysates could be performed. On page

17, the authors conclude that "the relative abundance of components of signaling pathways still vary by at least 5%". It should be ascertained that this lower limit is really biological and not technical.

The lower limit of variation we measure could be technical and variations may even be lower. We have now stated this more clearly in the manuscript. To assess this, we carried out experiments in which we measured the technical error in 30 identical lysates. This technical error is comparable to the biological variation. We now included these results in Figure EV3 and have added the following sentences to the text on p. 9:

"We would like to note that the biological variation might be for some proteins even lower than we were able to measure in these experiments. To test whether there is a lower limit for measuring variation, we carried out control experiments in which 30 individual eggs were lysed and mixed together to remove biological variability. This mixed lysate was then pipetted into 30 individual tubes, and the sample in each tube was prepared and analyzed separately by SRM mass spectrometry. The variation between these 30 individually prepared and analyzed aliquots of the same starting lysate were compared to obtain a measure of technical variation. As shown in Figure EV3, the technical variation is comparable to the lowest CV measurements we show in Figures 3A-C, suggesting that further technical improvements may reveal even lower biological variation. "

2. Figure 2B: show separate boxes for time point 60 and time point 80 min.

We have now separated the boxes for timepoint 60 and timepoint 80 min in the plot in Fig. 2C.

Figure 2C: To show that the two independent experiments show the same results, the same time points should be compared. Therefore, it would be better to show the CV of time point 60 min only on the y-axis.

We have now changed the figure panel to show the same timepoint from the 2 independent experiments.

3. Page 8, second paragraph, "we added covariation to the model": It is unclear how this covariation was added. Do the protein abundances perfectly correlate? In the results of Figure 3B, correlation was 0.53 and 0.38. More importantly, for the results in Figure 2D, the authors assume that the kinases OR the phosphatases covary together. As it is correctly pointed out on page 17, if the kinases covary with the phosphatases, the conclusions are completely different. Therefore, the impact of kinases AND phosphatases covarying together should also be modeled.

Yes, the MEK and ERK protein abundances do perfectly correlate when we add covariation to the model (we now add the noise terms the same way as in the other models). Also, we did choose to have 100% covariation of MEK and ERK in our model in order to maximize the possible effect of covariation. We tested that the effect is proportionally smaller if we reduce the degree of covariation. The measured value of covariation of MEK and ERK in MCF10A cells is 0.7, but this might be a lower limit due to experimental noise. We also state at different places in the manuscript that more components covarying increases the overall signal variation and that covariation of opposing enzymes can also cancel out the effect. We cite the latter in our manuscript as being shown previously for opposing enzymes in T-cell activation (Feinerman et al., 2008).

4. Figure 3C-D: Immunocytochemistry measurements have been performed, but only the analyzed data is shown. At least some examples of the acquired images should be shown.

We have now included examples of the acquired images in Figure EV4.

5. Page 12, second paragraph, "seven species in the MAPK model ... whose coefficient of variation were chosen to match the experimental data": Where is the experimental data for the seven species? Only the distributions of MEK and ERK were measured.

We have now clarified that in the data shown, we used 15% lognormal variation and 100% covariation for MEK and ERK to more clearly show the effect of covariation on the fraction of cells activated at increasing stimulus strength. We do not have experimental data for the other model parameters. Thus, we varied them with a 10% lognormal variation which is within the range of the variations we experimentally observed for different proteins in human cells.

6. Figure 4B: Why did the authors select such an unrealistically low ($r=0.04$) and high ($r=0.99$) correlation between MEK and ERK? In the experiments shown in Figure 3B and 3D, correlations of 0.38 and 0.62 have been measured for MEK and ERK, while for the uncorrelated proteins GAPDH and MCM5 a correlation of 0.15 was measured.

This is a good point that we now more clearly address in the paper. The measured value of MEK and ERK covariance in these MCF10A cells is 0.7 (data shown in new Figure 7F), but we used maximal (100%) covariation to better show how covariation increases the range of stimuli over which the fraction of activated cells is increasing. If we reduce covariation to smaller values, the effect is proportionally smaller. We have now stated this in the text on p.13:

"When comparing the effect of random variation versus covariation of MEK and ERK concentrations, we find a small but significant broadening of the relationship between the stimulus intensity and the fraction of cells in the active state when there is covariation (Figure 6B, orange versus blue curves). This effect is proportionally reduced if the covariation is partial and would increase if more pathway components - in addition to MEK and ERK - would covary with each other."

7. Page 13, first paragraph, "the correlated distribution had increased numbers of low responding cells and high-responding cells": In Figure 4D (paired), a peak at the highest possible integrated ERK activity can be seen. This presumably corresponds to an unrealistic complete ERK activation as seen in the plot with the highest ppERK level in Figure 5F. Such unrealistic simulations should be avoided. The same holds true for the high-frequency oscillations seen in Figure 5E and 5F.

This is a good point. We now show the effect of covariation by modeling and by experiments over a range of EGF doses to demonstrate that the effects we see are not only for unrealistic extremes of ERK activation.

8. Figure 4E-G: The authors claim that when MEK and ERK are correlated, the abundance of MEK correlates with the high responding population. An alternative explanation would be that only the abundance of ERK is predictive of a low versus high response and, if MEK and ERK are correlated, the same holds true for MEK. Therefore, the conclusion on page 13, second paragraph, "with covariation having a key role in controlling which cells are activated and which ones are not" is too strong.

We agree with the reviewer that our text and figures were not sufficiently clear and we adjusted how we describe the simulations and data. We now introduce covariance into the model as we did in the rest of the manuscript by setting the random log normal variation in MEK equal to that of ERK in each simulation (cell). It was not clear to us when we started the analysis of the model whether the concentration of MEK and ERK significantly matter as to whether a cell is activated or not. The simulations and also the experimental data show that variation in the concentration is a relevant factor for signal variability. In our analysis, variations in either MEK or ERK are both predictive of pathway activation (pERK) as now more clearly shown by how pERK increases (new Figure 6C).

9. Page 14, first paragraph, "the two proteins proved equally able to separate the two subpopulations of cells". According to the box plot in Figure 4J, there is no significant difference between the two subpopulations.

We have now clarified the plot to show that there is a significant shift of the two populations (new Figures 7D and 7E).

Minor comments:

1. Abstract and Page 2: The unit of protein expression variation ($\pm 25\%$) should be added, presumably CV is meant.

We are now more clearly stating that we are referring to a coefficient of variation of 25%.

2. Page 3, five lines from bottom, "to accurately population-level decisions": verb is missing.

We have now added the missing verb.

3. Page 5, seven lines from bottom, "an fIDL stimulus of 2.8": 2.83 would be better to match the figure.

We have now changed the text to say "2.83" to match the figure.

4. Page 7, second paragraph, "measurements in 25 individual *Xenopus laevis* eggs collected at 5 timepoints": please re-phrase to indicate that measurements have been performed at 5 time points with 5 eggs each.

We have rephrased this sentence.

5. Figure 1F: The number of dots of each timepoint is sometimes less than five. Are these missing values from the mass spectrometry experiments?

Thank you for noticing this. There are no missing values, but there had been an issue with the plotting script that we have now corrected.

6. Figure 2A: The median should be removed from the scatter plot as there are only five data points. However, the total median (7%) could be added as a line in the histogram.

We have replaced the median values with the mean values. Also, we have now left all the raw data as black points but have added a small red box to indicate the mean for each of the proteins. We feel it is

necessary to have a box marking the mean since It would be difficult at the final magnification to see what the mean variation is without a visual guide.

7. Figure 2B: Individual boxplots for $t = 60$ min and $t = 80$ min should be plotted. The median should be indicated with horizontal lines rather than with ovals.

We have now plotted individual boxes for timepoint 60min and timepoint 80min in the plot in Fig. 2B, and removed the ovals.

8. Page 9, second paragraph, "covariation between MEK and ERK in cultured human cells": It should be mentioned that these are HeLa cells.

We have now added to the text that these are HeLa cells.

Reviewer #3:

Kovary et al. present an analysis of the effect of protein expression variation on signal transduction pathways. Beginning with a model analysis, they motivate their study with simulations showing that the oft-cited 25% CV for expression variance shown in earlier studies would make it impossible for a 5-step signaling cascade to accurately relay stimulus levels. They then turn to experimental measurements of expression variance, first using *Xenopus* oocytes as a base case, and show a very nice set of measurements, with a median CV of 7%, arguing that low expression variance enabling more accurate signaling is possible within a biological system. They also observe co-variation between MEK and ERK expression levels, a finding they extend to cultured mammalian cells. They note using simulations that such co-variation can in fact enhance variance in signaling output, which is important in the case where the binary responses must be controlled at the population level, and they then test this hypothesis with an elegant experiment in which variable responses to a low dose of EGF, measured by live-cell reporter, are correlated with MEK and ERK expression levels in the same cell after fixation, providing support for the idea that the expression co-variance plays a part in the variable signaling response. Finally, they bridge the concepts of low variance as an important factor for accurate signaling at the single cell level and high variance as necessary for control of the fraction of cells undergoing a binary fate decision using a mathematical analysis of their cascade model.

Overall I found this manuscript thought provoking. It presents a nice exploration and synthesis of several ideas about the role of expression variance in signaling and some interesting experimental data that are in general supportive of the conceptual ideas. The mass spec study in particular seems very carefully performed. However, there are several points that should be addressed in revision:

1) The data in EV3 raise some important questions about the ERK/MEK immunofluorescence experiments. In the quantification of siRNA knockdowns, only a 30-40% reduction is seen for either MEK or ERK by siRNA. When this is presented as a validation of the antibody staining, it leaves open the question of whether the remaining signal is non-specific antibody staining or just incomplete knockdown. It is also not very convincing to have only a single siRNA for each gene. The conclusion that knockdown of MEK affects ERK and vice versa is questionable given the relatively small effect sizes and the well-known off-target effects of siRNA. The important points here could be made much stronger if additional knockdowns giving more complete depletion could be shown.

We have now found conditions in which MEK and ERK are significantly knocked down and have repeated the experiments showing the effects of MEK and ERK knockdown on each other's expression. Please see updated Figure EV5.

Also, why are MEK and ERK shown as bar graphs rather than histograms as for MCM5/7? If knockdown is not uniform across the population, that would certainly be relevant to the interpretation of the results.

We have now included histograms of MEK and ERK in Figure EV5.

2) The data in Fig. 4J are fairly impressive but, given their importance to the argument, need to be better controlled. It would be helpful to see one or more of the control stains (GAPDH, MCM5/7) to establish how an unrelated protein would behave in this experimental setup.

This is a very good point. In order to better control this experiment, we now normalized MEK and ERK in Fig. 4J (now Figure 7D and 7E) to the total protein stain used to normalize proteins in the other figures of the manuscript.

It would also be helpful (perhaps as a supplement) to see simulations such as in Fig. 4H/I with different degrees of randomization of MEK and ERK to help assess where the experimental result lies in the continuum between the two extremes of unsorted vs. fully sorted.

For both the MEK/ERK modeling and experiments, we now show the results for a range of input stimuli in order to better show the effect of covariance. In the figures of this manuscript, we used maximal (100%) covariation between MEK and ERK to better show how covariation increases the range of stimuli over which the fraction of activated cells can be regulated. If we reduce covariation to smaller values, the effect is proportionally smaller (the blue curve in Figure 6B will become more like the orange curve).

3) The ordering and grouping of the results might be reconsidered. While the writing is clear enough that I could follow the logic throughout, the narrative does feel a bit winding as it jumps from variance as a theoretical limit on signaling accuracy, to showing low variance in the oocytes, to the need for covariance at the population level. It seems likely that readers less familiar with the subject area would be confused by this. Contributing to this feeling is that the data in the first four figures are grouped a little oddly, with a new theme being introduced in the last few panels of each figure and then continued in more detail in the next figure.

In response to the reviewer's comments, we have reorganized the manuscript and rewritten key transition sections.

4) Various typos: "do to" instead of "due to" on page 9; "know" instead of "known" on page 7; missing word in "to accurately population level decisions" on page 3.

We have fixed the typos.

REFERENCES

Curran, K.L., and Grainger, R.M. (2000). Expression of Activated MAP Kinase in *Xenopus laevis* Embryos: Evaluating the Roles of FGF and Other Signaling Pathways in Early Induction and Patterning. *Dev. Biol.* *228*, 41–56.

Feinerman, O., Veiga, J., Dorfman, J.R., Germain, R.N., and Altan-Bonnet, G. (2008). Variability and robustness in T cell activation from regulated heterogeneity in protein levels. *Science* *321*, 1081–1084.

LaBonne, C., and Whitman, M. (1997). Localization of MAP kinase activity in early *Xenopus* embryos: implications for endogenous FGF signaling. *Dev Biol* *183*, 9–20.

Yang, H.W., Chung, M., Kudo, T., and Meyer, T. (2017). Competing memories of mitogen and p53 signalling control cell-cycle entry. *Nature* *549*, 404–408.

Thank you again for submitting your work to Molecular Systems Biology. We have now heard back from the referees who accepted to evaluate the revised study. They are now globally supportive and I am please to inform you that we will be able to accept your paper for publication in Molecular Systems Biology pending the following minor changes:

Remaining issues

- Reviewer #1 still raises an important concern with regard to the emphasis put of the role of covariation of ERK and MEK in generating a bimodal distribution. We have consulted with the other reviewers and reviewers who shared this opinion. Looking at figure 6B again, we also agree with this concern. We would thus ask you to remove this claim from the title, and considerably tone it down in the abstract and in the main text.

Main manuscript

- Please replace the manuscript PDF file with a MS Word file (LaTeX and RTF are also fine)
 - Please change the division of the manuscript parts from results and discussion followed by conclusions to just a results section followed by a separate discussions section.
<http://msb.embopress.org/authorguide#textformat>
 - Please change the references in the reference list from 10 authors + et al to 20 authors + et al to match the MSB reference style. <http://msb.embopress.org/authorguide#referencesformat>
 - Please remove the figure legends from the figure files. The legends should only appear in the manuscript file.

Callouts

- Please add a callout to figure EV4.
 - The callout to table EV5 appears before the callouts to table EV3 and EV4. Please update the callouts in numerical order.

Please supply in a separate file:

- three to four 'bullet points' highlighting the main findings of your study
 - a short 'blurb' text summarizing in two sentences the study (max. 250 characters)
 - a 'thumbnail image' (width=211 x height=157 pixels or, alternatively, 550px wide x 400px height max in Illustrator, PowerPoint, OmniGraffle or jpeg format), which can be used as 'visual title' for the synopsis section of your paper.

REVIEWER REPORTS

Reviewer #1:

Apart from technical issues, the main problem with the previous version of the manuscript was unclear presentation of the data. In the revised version of the manuscript, the authors did substantial work on the presentation of the results such that now these results are no longer obscured and can be interpreted by the readers. The data seem to support the hypotheses that variability in the concentration of ERK and MEK causes subpopulations of cells to respond/not respond. This is an important point. However, neither the data nor the simulations support the hypotheses that co-variation of ERK and MEK is really important to generate the observed bimodal distribution (analogue coding on the population level). Specifically, the simulations show that the main effect is due to variation of the individual proteins, and at best a small contribution comes from co-variation (there is hardly any difference between the two scenarios in Figure 7).

I find it hard to understand why the authors want to push their narrative while it is not supported. Why not simply state that 10%-15% CV is enough to generate signaling heterogeneity, and maybe

co-variation adds a bit more? The data is very nice and such quantification really necessary for the field, but the current discussion really weakens the paper.

Reviewer #2:

The authors answered all our questions and much improved the quality of the manuscript.

Reviewer #3:

The authors have significantly improved the manuscript, addressing essentially all of the reviewers' comments. This paper will contribute meaningfully to the analysis and understanding of variability in cell signaling pathways.

2nd Revision - authors' response

26 March 2018

Thank you again for submitting your work to Molecular Systems Biology. We have now heard back from the referees who accepted to evaluate the revised study. They are now globally supportive, and I am pleased to inform you that we will be able to accept your paper for publication in Molecular Systems Biology pending the following minor changes:

Remaining issues

- Reviewer #1 still raises an important concern with regard to the emphasis put of the role of covariation of ERK and MEK in generating a bimodal distribution. We have consulted with the other reviewers and reviewers who shared this opinion. Looking at figure 6B again, we also agree with this concern. We would thus ask you to remove this claim from the title, and considerably tone it down in the abstract and in the main text.

We have now reorganized the figures on variation and covariation (previous Figures 5D, 6, and 7) and added additional analysis. We first focus on variation alone in Figures 5 and 6 and then more clearly show the role covariation of MEK and ERK has in generating a bimodal distribution only in Figure 7. In particular, we have demonstrated with the lower panel in Figure 7C that covariation allows for better control of the fraction of activated cells more narrowly for low ranges of receptor stimuli.

We have also toned down the abstract and main text to emphasize that the main principle is centered on variation, but that increased variation, covariation, and number of components are all effective mechanisms to increase system noise and thus controllability of the number of cells in a population that make a binary fate decision.

The abstract now states:

“Focusing on bimodal ERK signaling, we show that variation and covariation in MEK and ERK expression improves controllability of the fraction of activated cells, demonstrating how variation and covariation in expression enables population-level control of binary cell-fate decisions. Together, our study argues for a control principle whereby low expression variation enables accurate control of analog single-cell signaling, while increased variation, covariation, and numbers of pathway components are required to widen the stimulus-range over which external inputs regulate binary cell activation to enable precise control of the fraction of activated cells in a population.”

We have also included in the Discussion text which tones down the effect of covariation of just a single pair of proteins such as MEK and ERK:

“Of note, the contribution from a single pair of co-varying signaling proteins is relatively small, and strong effects resulting from co-variation require multiple signaling proteins co-varying with each other.”

Main manuscript

- Please replace the manuscript PDF file with a MS Word file (LaTeX and RTF are also fine)
 - Please change the division of the manuscript parts from results and discussion followed by conclusions to just a results section followed by a separate discussions section. <http://msb.embopress.org/authorguide#textformat>
 - Please change the references in the reference list from 10 authors + et al to 20 authors + et al to match the MSB reference style. <http://msb.embopress.org/authorguide#referencesformat>
 - Please remove the figure legends from the figure files. The legends should only appear in the manuscript file.
- We have now made all these changes to the manuscript.

Callouts

- Please add a callout to figure EV4.
We have added a callout to Figure EV4 on p. XXX.
- The callout to table EV5 appears before the callouts to table EV3 and EV4. Please update the callouts in numerical order.
We have corrected the callouts to the tables.

Please supply in a separate file:

- three to four 'bullet points' highlighting the main findings of your study
 - a short 'blurb' text summarizing in two sentences the study (max. 250 characters)
 - a 'thumbnail image' (width=211 x height=157 pixels or, alternatively, 550px wide x 400px height max in Illustrator, PowerPoint, OmniGraffle or jpeg format), which can be used as 'visual title' for the synopsis section of your paper.
- The thumbnail image has been uploaded as a separate file. However, we have included the bullet points and blurb text at the end of this document since we did not see a place to upload these as a separate file on your submission website.

Reviewer #1:

Apart from technical issues, the main problem with the previous version of the manuscript was unclear presentation of the data. In the revised version of the manuscript, the authors did substantial work on the presentation of the results such that now these results are no longer obscured and can be interpreted by the readers. The data seem to support the hypotheses that variability in the concentration of ERK and MEK causes subpopulations of cells to respond/not respond. This is an important point. However, neither the data nor the simulations support the hypotheses that co-variation of ERK and MEK is really important to generate the observed bimodal distribution (analogue coding on the population level). Specifically, the simulations show that the main effect is due to variation of the individual proteins, and at best a small contribution comes from co-variation (there is hardly any difference between the two scenarios in Figure 7).

I find it hard to understand why the authors want to push their narrative while it is not supported. Why not simply state that 10%-15% CV is enough to generate signaling heterogeneity, and maybe co-variation adds a bit more? The data is very nice and such quantification really necessary for the field, but the current discussion really weakens the paper.

Please see our response above to the editor.

Reviewer #2:

The authors answered all our questions and much improved the quality of the manuscript.
We thank the reviewer for his/her positive comments.

Reviewer #3:

The authors have significantly improved the manuscript, addressing essentially all of the reviewers' comments. This paper will contribute meaningfully to the analysis and understanding of variability in cell signaling pathways.
We thank the reviewer for his/her positive comments.

Corresponding Author Name: Mary Teruel

Manuscript Number: MSB-17-7997